# From sabers to spikes: A newfangled reconstruction of the ancient, giant, sexually dimorphic Pacific salmon, †*Oncorhynchus rastrosus* (SALMONINAE: SALMONINI)

**Kerin M. Claeson**[1]*, **Brian L. Sidlauskas**[2☯], **Ray Troll**[3☯], **Zabrina M. Prescott**[4☯], **Edward B. Davis**[5☯]

**1** Department of Bio-Medical Sciences, Philadelphia College of Osteopathic Medicine, Philadelphia, Pennsylvania, United States of America, **2** Department of Fisheries, Wildlife and Conservation Sciences, Oregon State University, Corvallis, Oregon, United States of America, **3** Troll Art Studios, Ketchikan, Alaska, United States of America, **4** Department of Biology, Dalhousie University, Halifax, Nova Scotia, Canada, **5** Museum of Natural and Cultural History and Department of Earth Sciences, University of Oregon, Eugene, Oregon, United States of America

☯ These authors contributed equally to this work.
* kerincl@pcom.edu

**Data Availability Statement:** The entire data matrix is retrievable with additional supporting images at MorphoBank.org, Project 2734, http://

## Abstract

The impressive †*Oncorhynchus rastrosus* of the Pacific Northwest's Miocene and Pliocene eras was the largest salmonid ever to live. It sported a hypertrophied premaxilla with a pair of enlarged teeth which the original describers reconstructed as projecting ventrally into the mouth, leading them to assign the species to "*Smilodonichthys*," a genus now in synonymy. Through CT reconstruction of the holotype and newly collected specimens, we demonstrate that the famed teeth projected laterally like tusks, not ventrally like sabers or fangs. We also expand the original description to characterize sexual dimorphism in mature, breeding individuals. Male and female †*Oncorhynchus rastrosus* differ in the form of the vomer, rostro-dermethmoid-supraethmoid, and dentary, much as do other extant species of *Oncorhynchus*. Male specimens possess a more elongate vomer than do females, and female vomers have concave ventral surfaces and prominent median dorsal keels. The dentary of females has no evidence of a kype, though some specimens of †*O. rastrosus* have a non-uniform density mesial to the tooth bed, which we interpret as a male kype. Unlike extant *Oncorhynchus*, male and female †*O. rastrosus* do not differ in premaxilla shape. Because male and females possess hypertrophied premaxillae and lateral premaxillary spikes, the former common name "Sabertoothed Salmon" no longer reflects our understanding of the species' morphology. Accordingly, we redub †*O. rastrosus* the Spike-Toothed Salmon and postulate that its spikes were multifunctional, serving as defense against predators, in agonism against conspecifics, and as a practical aid to nest construction.

dx.doi.org/10.7934/P2734 A labeled surface model of the female skull is available on SketchFab at this link: https://sketchfab.com/3d-models/spike-toothed-salmon-female-skull-model-7b1c9ddcd3f04cfb8a321ebd34390d92.

**Funding:** This work was supported by the National Science Foundation (1948340 and 2228394, awarded to ED). The funders had no role in study design, data collection and analysis, decision to publish, or preparation of the manuscript.

**Competing interests:** The authors have declared that no competing interests exist.

## Introduction

Trouts and salmons (*Oncorhynchus*: SALMONINAE), captivate human interest like few other fishes on the planet. Given their large size, lengthy migrations, powerful swimming abilities, distinctive appearance and appetizing flesh, it is not surprising that they have supported countless sport and commercial fisheries worldwide [1], formed the subject of thousands of scientific investigations or conservation interventions (e.g., Google Scholar search on "salmon" July 31, 2023 yielded 2,480,000 results), and figured prominently in human myth, ceremony, and art for millennia. For example, the Irish Fenian cycle tells the tale of the Salmon of Knowledge [2], salmon rituals and tales of the "divine fish" figure prominently in the folklore of Japan's Ainu people [3] (Obayashi, 1996), and salmon motifs appear ubiquitously in the artwork of the tribes native to North America's Pacific Northwest [4] (Leung, 2006). Furthermore, as recently as 1974, treaty-guaranteed fishing rights were restored to Puget Sound Native Americans and a renewal of the First Salmon Ceremony marked a recognition of the association with salmon and the special role Native Americans play in preserving this economic and spiritual resource for all people [5] (Amoss, 1987).

In the Pacific Northwest, *Oncorhynchus* is the most celebrated and species-rich salmonine subclade, yet the twelve living species represent only a portion of their clade's historical diversity. For example, migratory anadromous salmonines of Miocene and Pliocene rivers and lakes provide evidence for drainage connections to the Pacific Ocean [6–8]. Several of the extinct species (†*Oncorhynchus salax*, †*O. ketopsis*) represented in the Late Cenozoic fossil record of the western United States [9] display morphologies unrepresented among the modern fauna, such as hyperelongate gill rakers that apparently adapted species like *O. salax* to an exclusively filter feeding niche [9–11].

Among these salmons of the past, none have attracted more fame than the enormous "Sabertoothed" Salmon, †*Oncorhynchus rastrosus* (Fig 1A), known primarily from its holotype, UOMNH F-26799, and paratype, UOMNH F-3335, which were recovered from the freshwater Gateway Locality of the Madras Formation, near the town of Gateway, Jefferson County, Oregon [6] (Cavender & Miller, 1972). Coastal marine and freshwater sediments from California have yielded additional specimens of this species, and isotope analysis clearly supports the conclusion that this giant species was anadromous [11] (Koch, et al., 1992), much like the largest living member of its genus, *Oncorhynchus tshawytscha*, the King Salmon.

This giant extinct species reached between 2.4–2.7 meters in length by some estimations [6, 9], making it the largest known salmonid ever to live, substantially exceeding the largest living species, the Siberian Taimen *Hucho taimen*, which grows to about two meters [12] (Kottelat & Freyhof, 2007). While its large size alone would have rendered †*Oncorhynchus rastrosus* memorable, it also bore a greatly enlarged tooth on each premaxilla. That unusual morphology appeared not only on the holotype and paratype, but also on isolated premaxillae found in other regions of Oregon and California [6] (Cavender & Miller, 1972). Morphological variation of the premaxillae and osseous bases of the premaxillary teeth of †*O. rastrosus* further supports the conclusion that this species was anadromous, in that these structures are larger in freshwater fossils than in marine specimens. Therefore, the increase in tooth size indicates a developmental change associated with spawning migration [8] (Sankey, et al., 2016).

Initially, all recovered premaxillae were disassociated from the remainder of the skull, thus, the fossil's discoverers reconstructed this tooth as projecting ventrally into the oral cavity as is in other salmonines (Fig 1A). A configuration reminiscent of a sabertoothed tiger resulted, which led those authors to assign the distinctive species referentially to the new genus †*Smilodonichthys*. A later phylogenetic study [7] (Stearley & Smith, Phylogeny of the Pacific Trouts and Salmons (*Oncorhynchus*) and Genera of the Family Salmonidae, 1993) recognized

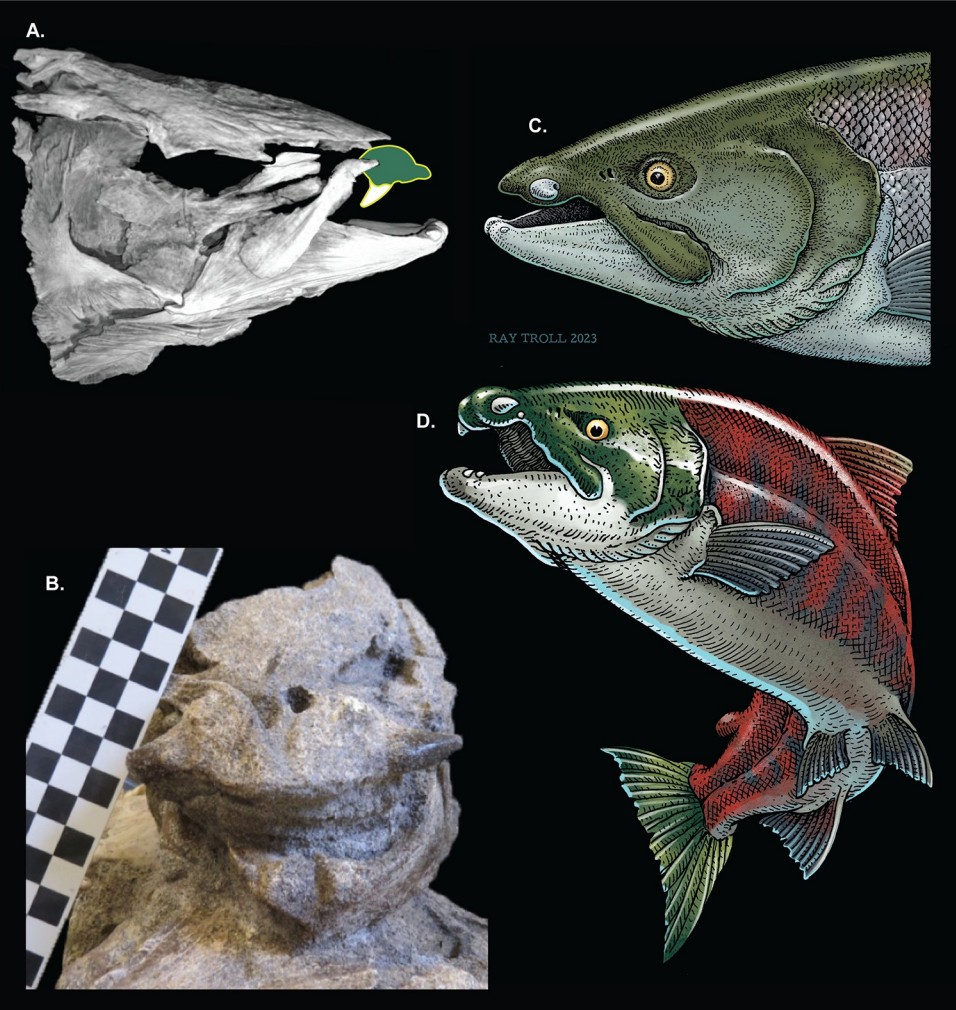

**Fig 1. †*Oncorhynchus rastrosus*.** (A) CT model of Holotype, UO F-26799, skull in right lateral view with a stylized drawing of the originally proposed "sabertoothed" position of the isolated premaxilla; (B) UO_A in anterior view of skull, prior to complete preparation and CT scan; (C) Artist's rendering skull of male iconic fish with accurate spike-tooth configuration; (D) Artist's rendering of complete female iconic fish with accurate spike-tooth configuration. Scale bar blocks = 1 cm each.

numerous similarities between the fossils and the living species *Oncorhynchus nerka* (Sockeye Salmon), most notably the possession of the numerous filamentous gill rakers that allow *O. nerka* to feed effectively on plankton. Because of that and several other clear synapomorphies uniting the Sabertoothed and Sockeye Salmons, Stearley and Smith reconstructed these species as belonging to a deeply nested clade also including *O. gorbuscha* and *O. keta* and recognized †*Smilodonichthys* as a junior synonym of the (admittedly more pedestrian) *Oncorhynchus* [7] (Stearley & Smith, Phylogeny of the Pacific Trouts and Salmons (*Oncorhynchus*) and Genera of the Family Salmonidae, 1993).

A 2014 expedition led by EBD to the Gateway Locality recovered additional skull and post-cranial material. Later the North America Research Group later donated a specimen from the same locality to the University of Oregon Museum of Natural History (UOMNH). These exceptionally well-preserved specimens included several skulls with articulated premaxillae, revealing that the prominent teeth projected laterally like spiked tusks, rather than ventrally

like fangs (Fig 1B). That fundamental difference with the earlier reconstruction prompted our full reexamination of †*O. rastrosus*. In this report, we present an amended systematic paleontology, detail the distinctive morphology of the species via CT scanning, and illustrate the new reconstruction of this iconic animal (Fig 1C). We also test the hypothesis that †*O. rastrosus* is a derived member of Salmoninae using parsimony and Bayesian methods while including additional extinct filter feeding salmons in the analysis.

## Materials and methods

### Specimens examined

No permits were required for the described study, which complied with all relevant regulations.

Fossils: *Oncorhynchus nerka*, UWBM-87112, UWBM-87121, UWBM-87122, UWBM-95842; †*Oncorhynchus rastrosus*, UOMNH F-26799 holotype, UOMNH F-3335 paratype, UO F-55101 A-F, UWBM71908a, UWBM71908b, UWBM50816; Salvelinus sp. UO F-26785, UO F-26786, UO F-26787.

Modern comparisons: *Hucho hucho* UW 022130, ZIN 7872; *Oncorhynchus clarkii* UW 028776, CMNFI-1973-0393.1; *Oncorhynchus gorbuscha*, UW 16027, CMNFI-1984-0169.1; *Oncorhynchus keta*, UW 13664; *Oncorhynchus kisutch*, UW 015091, CMNFI-Z000804; *Oncorhynchus mykiss*, UW 016073; *Oncorhynchus nerka*, UW 005624, CMNFI-1977-0277.1. *Salmo salar* UW 019807, CMNFI-1980-0181.1; *Salvelinus aplinus*, UW 041192, CMNFI-1961-0229, CMNFI-1968-1262.1, CMNFI-1977-0348.1, CMNFI-1979-1001.1; *Salvelinus confluentus*, UW 20760; *Salvelinus fontinalis* UW 28888, CMFI-1973-0255.1; *Salvelinus leucomaenis*, UW 042437; *Salvelinus namaycush*, UW 020761, CMNFI-1982-0385, CMNFI-1958-0100A.1

### Excavation and preparation

The new specimens were recovered from the Gateway Quarry at the same stratigraphic level as the holotype and paratype described by [6]. The productive layer of the quarry falls at a stratigraphic transition between fine-grained, finely-bedded sediments and a poorly-sorted conglomerate that includes clasts ranging from <1cm to ~30 cm in diameter. The quarry is located on private land, and the UO crew is grateful to the landowners for permission to collect and for generously donating the specimen to the UO Museum of Natural and Cultural History.

Initial excavations in 2011 had revealed the presence of two individuals of †*O. rastrosus* in the quarry, but the overhanging conglomerate was too dangerous to allow the excavation to proceed at that time. After three years of natural weathering, the site was safe for excavation, so the UO crew spent one day uncovering the specimens in a single block, jacketing them with plaster, and removing them to the museum. Unfortunately, the specimens were damaged on the way from the quarry to the field vehicle. Much of this damage was repaired during preparation.

Upon return to the museum, the specimens were prepared using hand tools and consolidated using B-72 adhesive following standard techniques. The two skulls (Figs 2 and 3) were uncovered directly contacting one another, at right angles. They were carefully separated and are now stored in separate cradles. All specimens from this excavation were given a single specimen number because of the difficulty in ascertaining which skeletal elements might have belonged to which individual.

### CT scanning and digital reconstruction

The holotype and new specimens were scanned at the Oregon Imaging Centers on a Philips Brilliance 64 medical CT scanner at Oregon Imaging Centers in Eugene, Oregon. The scans

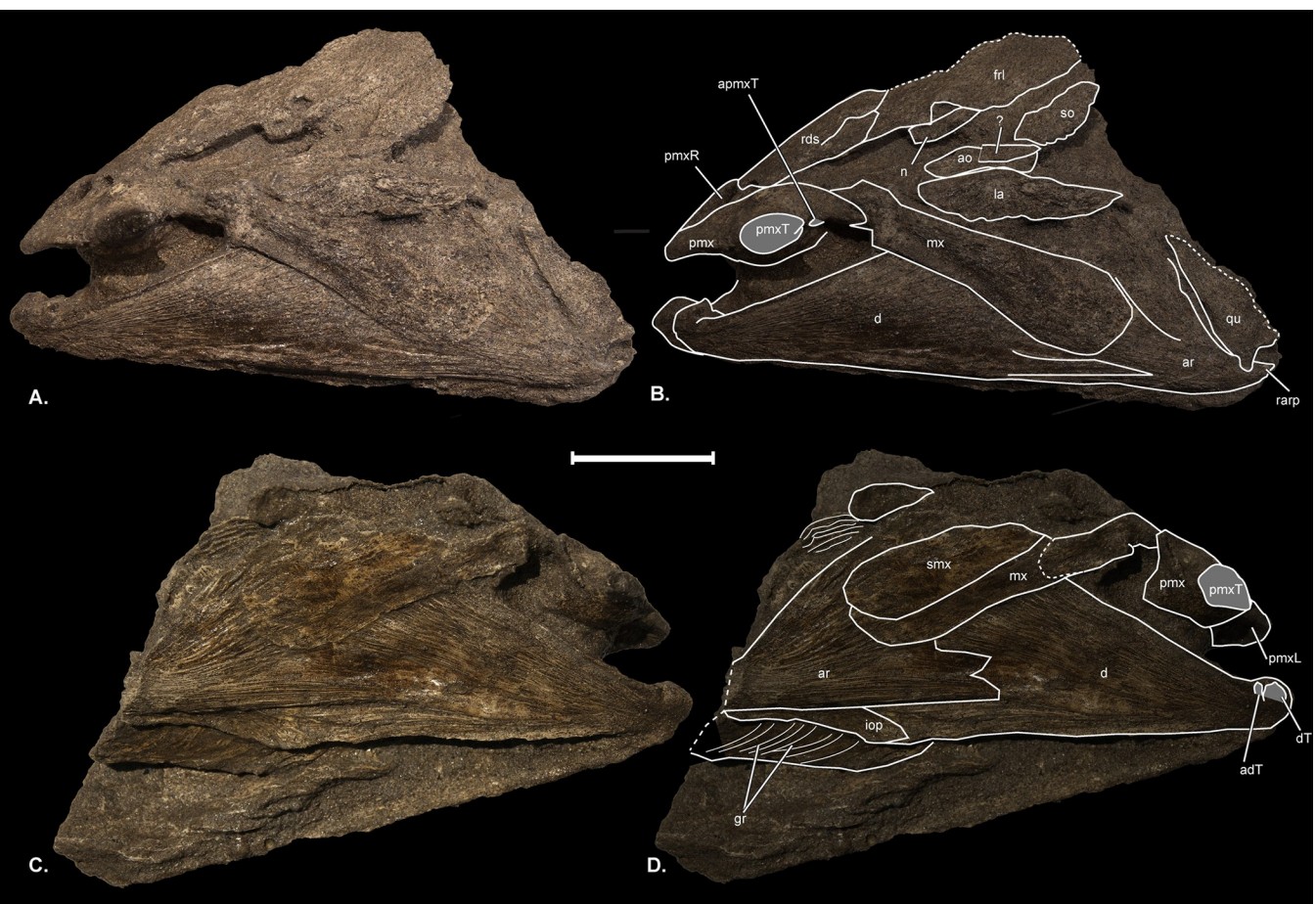

**Fig 2. †*Oncorhynchus rastrosus* female skull, UO F-55101, specimen A.** (A) in left lateral view with (B) line drawings overlain; (C) in right lateral view with (D) line drawings overlain. Scale bar = 5 cm. Full page, turned landscape.

were performed at an energy of 140 kV and an intensity of 997 μA. A filter detail was used to reduce noise, and some orthogonal reconstructions were done post-scan. Resulting DICOM datasets were analyzed using VGStudio Max v2.1, v3.1, and OsiriX v4.1 at Philadelphia College of Osteopathic Medicine. Additional comparative materials were scanned at the National Research Council Canada Biomedical MRI Research Lab (Halifax, NS) with a Lab PET 4 machine and a Triumph X-O CT as part of the thesis research of co-author ZMP [13]. The pre-maxillae (which bear the famous teeth) and remaining skeletal elements were digitally prepared away from the remainder of the skull and matrix using VG Studiomax to allow direct comparison with isolated premaxillae of the holotype and of growth series.

## Terminology

Osteological nomenclature generally follows Stearley and Smith [7] (Stearley & Smith, Phylogeny of the Pacific Trouts and Salmons (*Oncorhynchus*) and Genera of the Family Salmonidae, 1993). In accordance with Cavender and Miller [6] (Cavender & Miller, 1972), we use endopterygoid (= mesopterygoid of [7] (Stearley & Smith, Phylogeny of the Pacific Trouts and Salmons (*Oncorhynchus*) and Genera of the Family Salmonidae, 1993)) to refer to the plate-like bone of the suspensorium that is anterior to the quadrate and inferior to the orbit. In accordance with Sanford [14] and Patterson [15], we use rostro-dermethmoid-supraethmoid

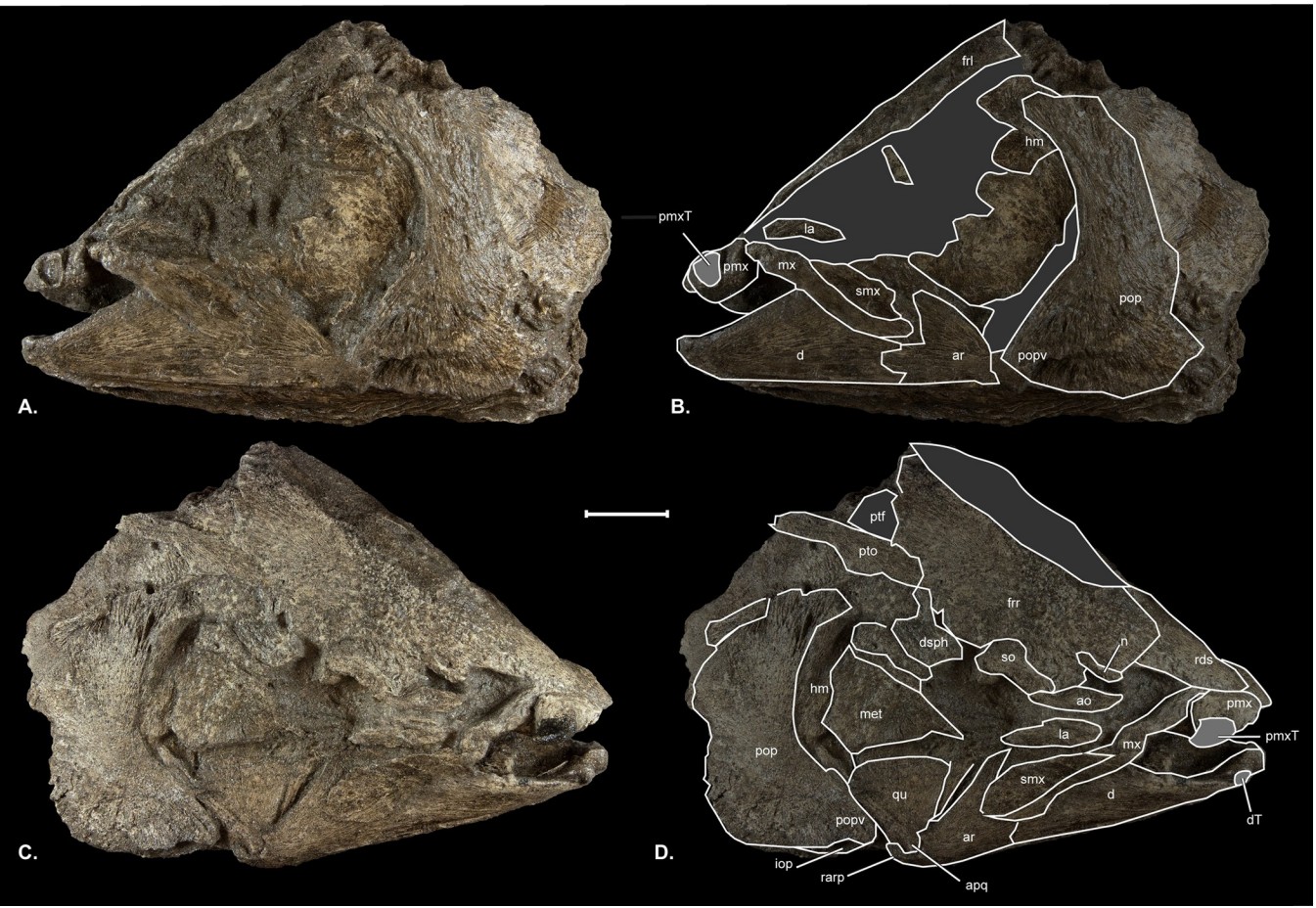

**Fig 3. †*Oncorhynchus rastrosus* skull, male, UO F-55101, specimen B.** (A) in left lateral view with (B) line drawings overlain; (C) in right lateral view with (D) line drawings overlain. Scale bar = 5 cm. Full page, turned landscape.

(abbreviated rds) to indicate the dermal bone medial to the paired premaxilla. This ossification is synonymous with the supraethmoid [6] and the dermethmoid [7].

## Anatomical abbreviations

adT = accessory dentary tooth; ao = antorbital; apmxT = accessory premaxillary tooth; apq = articular process of the quadrate; ar = articular; aT = accessory tooth; bh = basihyal; br = branchiostegal; ch = ceratohyal; chl = left ceratohyal; chr = right ceratohyal; chp = posterior ceratohyal; cl = cleithrum; co = coracoid; d = dentary; dsph = dermosphenotic; dT = dentary teeth; ect = ectopterygoid; eh = epihyal; end = endopterygoid; fr = frontal; frl = left frontal; frr = right frontal; ga = gill arches; gr = gill rakers; hem = hemitrichia; hh = hypohyal; hhl = left hypohyal; hhr = right hypohyal; io = infraorbital; iop = interopercle; mpt = metapterygoid; mx = maxilla; n = nasal; na = neural arch; ns = neural spine; pc = postcleithrum; pcl = left postcleithrum; pcr = right postcleithrum; pmx = premaxilla; pmxf = medial premaxillary flange to ethmoid cartilage; pmxT = premaxillary tooth (spike); pmxw = dorsal premaxillary wing to maxilla; pop = preopercle; popd = dorsal limb of preopercle; popv = ventral limb of preopercle; ppq = posterior process of the quadrate; pro = prootic; prr = proximal radials; ps = parasphenoid; pt = posttemporal; ptf = posttemporal fossa; q = quadrate; rarp = retroarticular process; rds = rostro-dermethmoid-supraethmoid;

sc = scapula; scl = supracleithrum; smx = supramaxilla; so = supraorbital; soc = supraorbital canal; sy = symplectic; uh = urohyal; v = vomer; vp = ventral V-shaped peak of posterior articular surface of proatlas and anterior articular surface of first free centrum.

### Institutional and other abbreviations

CMNFI = Canadian Museum of Nature Fish collection; UMMP = University of Michigan Museum of Paleontology; UOMNH = University of Oregon Museum of Natural and Cultural History; UO_A = UO F-55101, specimen A; UO_B = UO F-55101, specimen B; UO_C = UO F-55101, specimen C; UO_D = UO F-55101, specimen D; UO_E = UO F-55101, specimen E; UO_F = UO F-55101, specimen F; UWBM = Burke Museum of Natural History and Culture; University of Washington Fisheries collections (Seattle, Washington, USA); † = the dagger symbol indicates the taxon is extinct.

### Phylogenetic character analysis

We use species as our operational taxonomic unit (OTU) throughout this analysis. The data matrix includes the original 119 characters and 38 OTUs of Stearley and Smith [7]. To that matrix, we added previously missing data pertaining to the visceral skeleton and basicranium of †*Oncorhynchus rastrosus* for characters newly visible on the recently collected specimens. We also added the extinct filter feeding taxa, †*Oncorhynchus salax*, †*O. ketopsis*, and †*O. rastellus*. Twenty-nine of the 42 OTUs in the resulting matrix are salmonine ingroups and include all living species of that clade, while 12 are outgroup taxa. Characters were scored as missing data (?) when it was not possible to observe character states owing to preservation or preparation limitations on specimens. Characters were scored as not-applicable (-) when it was not possible to score character states owing to the absence of homologous structures (i.e., in the absence of a structure, qualities of that structure cannot be addressed). The data matrix was compiled using Mesquite v3.61 [16]. Morphology is illustrated for †*O. rastrosus* in the figures accompanying this paper and characters are annotated in Morphobank.org [17].

The data matrix was analyzed using parsimony and Bayesian approaches. We conducted the parsimony analysis with PAUP* 4.0b10 [18] (Swofford, 2002) using the maximum parsimony optimality criterion. Multistate scores were treated as polymorphic. We employed heuristic searches with 1000 replicates of random stepwise addition (branch swapping: tree-bisection-reconnection) holding one tree at each step and implementing a branch-and-bound search. Branches were collapsed to create soft polytomies if the minimum branch length was equal to zero (amb- option). Bremer support [19] (Bremer, 1994) values for nodes in all most parsimonious trees in PAUP* were calculated using constraint trees generated in MacClade [20] (Maddison & Maddison, 2000) from the 'Decay Index PAUP* File' command. Character states were optimized using ACCTRAN and DELTRAN algorithms to evaluate character consistencies. Unambiguous optimizations were recovered for all nodes retained in the consensus tree as reported by MacClade. The Bayesian analysis was conducted in MrBayes [21] (Ronquist & Huelsenbeck, 2003). Parameters are available in the S1 Matrix.

### Systematic paleontology

Teleostei Patterson and Rosen, 1977 [22]

Salmoniformes Johnson and Patterson, 1996 [23]

Salmonidae Jarocki 1822 or Schinz 1822 [24]

Salmoninae Jarocki or Schinz 1822

*Oncorhynchus* Suckley 1861 [25]

Type species—*Salmo gorbuscha* Walbaum 1792 [26]

1792 *Salmo gorbuscha*, Walbaum p. 69

1836 *Salmo scouleri*, Richardson, [27] p. 158, pl. 93

1861 *Oncorhynchus scouleri*, Suckley, [25] p. 313

†*Oncorhynchus rastrosus* (Cavender and Miller, 1972) [6] (Cavender & Miller, 1972)

Figs 1–9

1972 *Smilodonichthys rastrosus*, Cavender and Miller, [6] p. 4, Figs 3–13

1993 *Oncorhynchus rastrosus*, Stearley and Smith, [7] p. 23, Fig 12

Type species—*Smilodonichthys rastrosus*[6]dimorphic character

## Referred material

Holotype, UO F-26799 (male); paratype UO F-3335; UMMP V58061 vertebrae belonging to the paratype; UWBM-50816; UWBM-71908a; UWBM-71908b; new specimens UO F-55101 A-F. UO_A (UO F-55101 skull—*female*), UO_B (UO F-55101 skull—*male*), UO_C (UO F-55101 vertebrae with dorsal fins, opercle, and pectoral girdle), UO_D (UO F-55101 abdominal vertebrae), UO_E (UO F-55101 postcranium), UO_F (UO F-55101 left articular isolated). Skulls were found perpendicular to one another *in situ*. Vertebrae were partially weathered and adjacent to skulls. Based on CT reconstruction, we consider UO_A and UO_C to be associated crania and postcrania. It is unclear to which skull UO_E belonged.

## Locality and age

Gateway Locality of the Madras Formation, near the town of Gateway, Jefferson County, Oregon.

## Emended diagnosis

Species of *Oncorhynchus* unlike all other species of that genus in possessing the following combination of characters: laterally directed premaxillary and dentary teeth (versus ventrally directed premaxillary teeth and dorsally directed dentary teeth), maxilla edentulate (versus toothed), over 100 gill rakers on first branchial arch (versus fewer than 60 such rakers in all other known species), and first centrum fused to basioccipital forming simple occipital condyle (versus first centrum not fused to basioccipital, and occipital condyle formed by exoccipitals and basioccipital). Most similar to *O. rastellus*, sharing massive tooth on premaxilla, one or more small teeth on dentary nearest symphysis, more open form of the angular-articular condyle than in any other members of *Oncorhynchus*, and fusion of the first vertebral centrum with the basioccipital; separated from *O. rastellus* by possessing accessory tooth posterior to massive premaxillary tooth (versus absence of such a tooth), lacking maxillary teeth (versus possessing two such teeth), and by massive maximum size of 2.7 meters (versus maximum known size of 0.5 meters).

## Description

**Skull.**   The newly available skulls include fully articulated jaw bones that are preserved in three-dimensions (Figs 1–5). The median **rostro-dermethmoid-supraethmoid bone (rds)** in UO_A, UO_B, and the holotype is triangular in dorsal view with the maximum length along the anteroposterior axis approximately equal to the bone's width (Figs 4 and 5). In UO_A the rds is blunt rostrally (Fig 4C and 4D), unlike the more acute morphology of the holotype (Fig 4E and 4F) and UO_B (Fig 4A and 4B). Posteriorly, a notch in the rds of UO_A and UO_B forms the anterior portion of a large cartilaginous interspace separating the contralateral frontals and thereby creates posterolateral wings of the rds. The right posterolateral wing

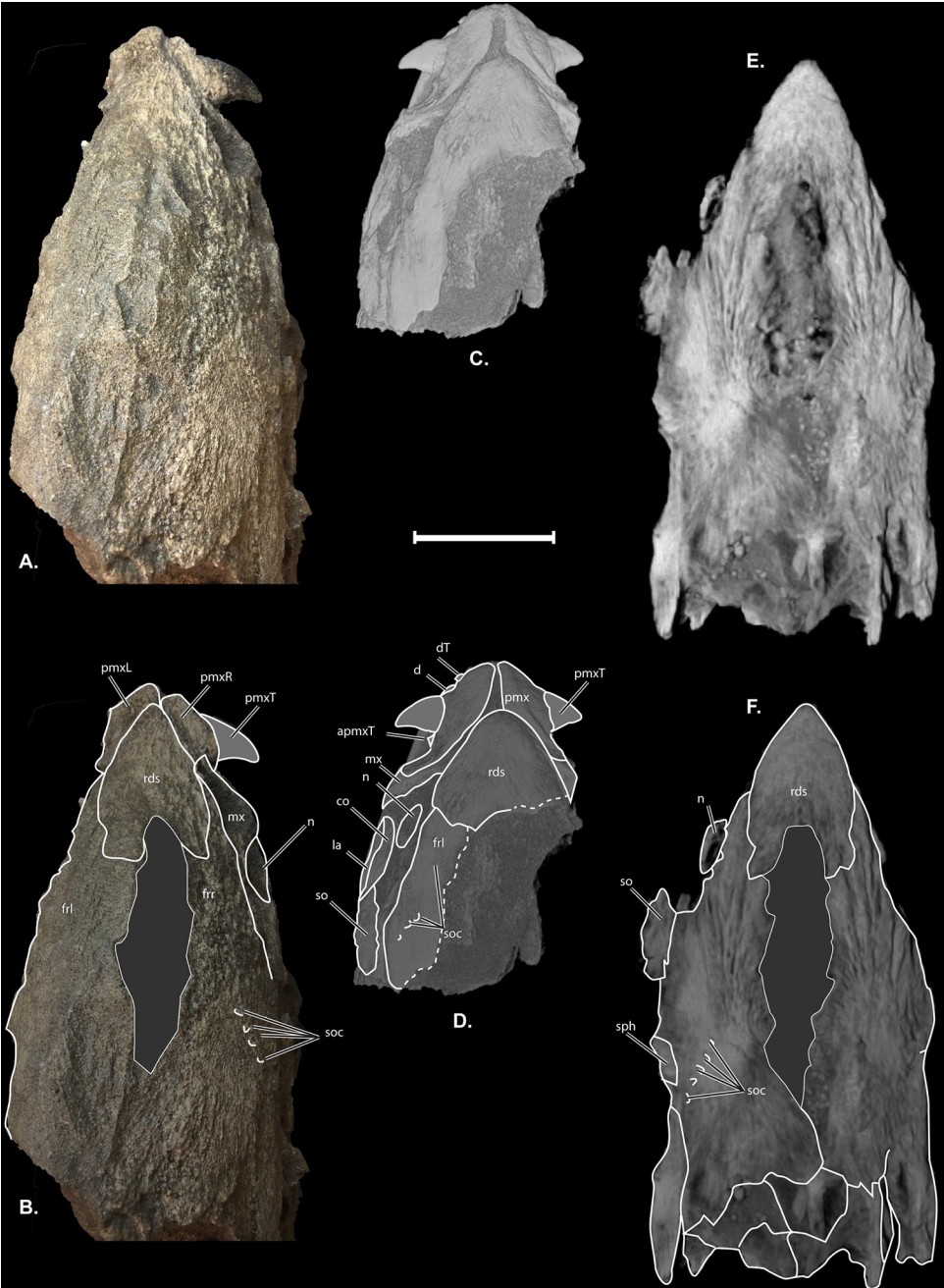

**Fig 4. †*Oncorhynchus rastrosus* skulls in dorsal view.** (A) UO F-55101, specimen B = male with (B) line drawings overlain; (C) UO F-55101, specimen A = female with (D) line drawings overlain; (E) CT rendering of Holotype UO F-26799 = male with (F) line drawings overlain. Anterior to top of page. Scale bar = 5 cm. Full page, portrait.

of the rds in UO_A is not preserved. The CT scan of the holotype confirms a notched rds and the presence of wings that are relatively shorter than those in UO_A.

The skull roof of UO_A preserves an incomplete **left frontal (frl),** though the **right frontal bone (frr)** has been lost (Figs 2, 4, and 5). UO_B possesses more completely preserved frontals (Figs 3 and 4) that form a broad anterior orbital shelf clearly visible on both sides of the specimens. The anterior aspect of each frontal articulates with a posterolateral wing of the rds. The

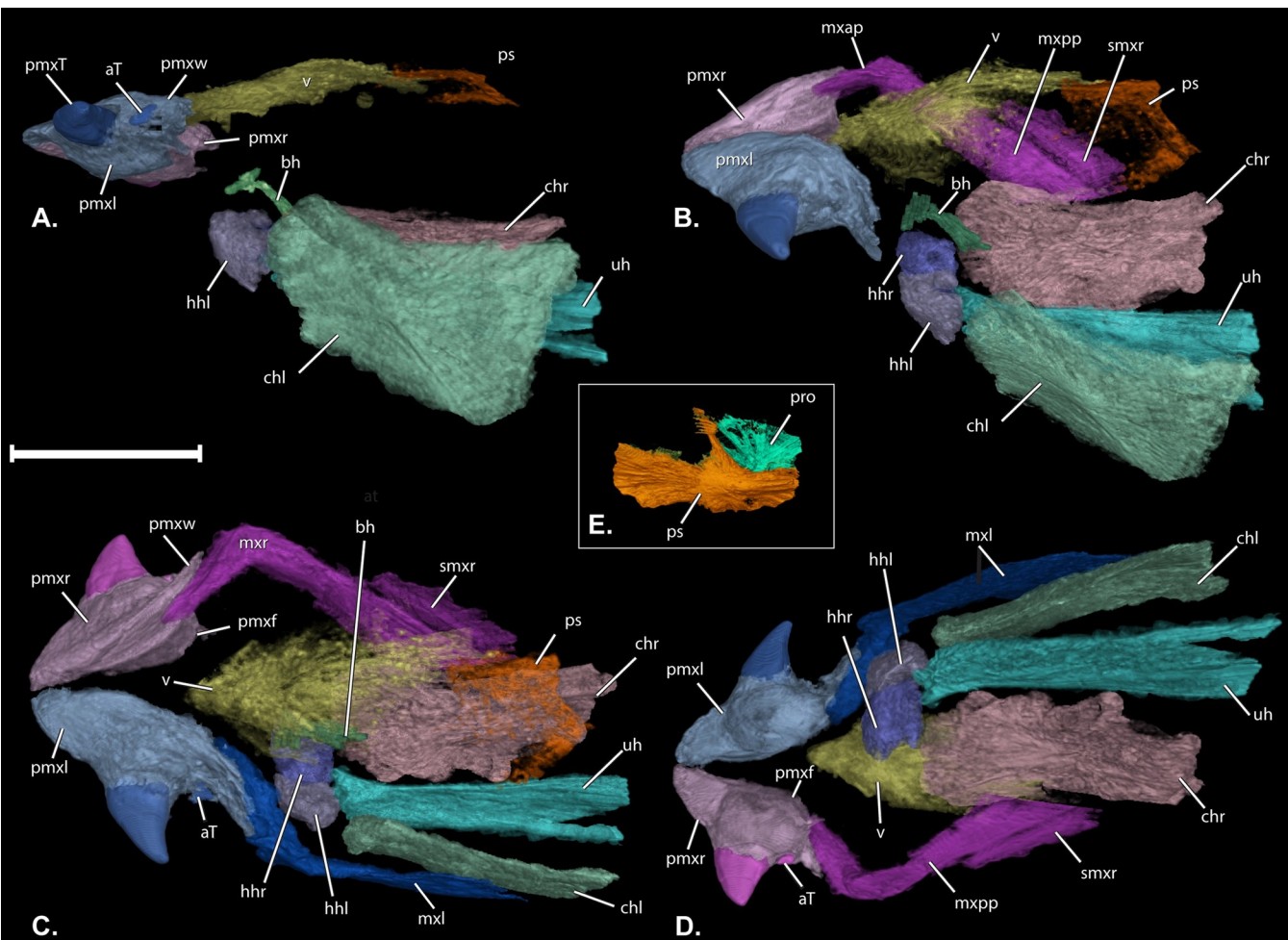

**Fig 5. Segmented skull of †*Oncorhynchus rastrosus*, UO F-55101, specimen A.** (A) left lateral view; (B) left dorso-lateral oblique view; (C) dorsal view; (D) ventral view; (E) isolated parasphenoid and prootic bones in ventral view. Anterior to left of page. Scale bar = 5 cm. Full page, turned landscape.

frontals do not articulate with each other medially, and the resulting space on the midline of the skull is interpreted as a cartilaginous interspace rather than a truly open fontanelle, as that region is filled with cartilage in modern salmonids (4). The **supraorbital canals (soc)** are best seen on the frontal bone in the CT scan of UO_A and the holotype (Fig 4D and 4F). Anterolateral to the anteriormost visible canal pore of UO_A is a small **nasal bone (n)**. Posteriorly on the right side of UO_B (Fig 3B), there is an incomplete, shallow **posttemporal fossa (ptf)**. The ptf is not preserved on the left side of UO_B. Neither pterotic of UO_A or UO_B was preserved.

The anterior circumorbital bones are preserved on the left side of UO_A (Fig 2A and 2B). A small, ovoid **infraorbital bone (io)** overlaps a larger, ovoid **lacrimal (la)** to a greater degree in UO_A than it does in the holotype. Superiorly, the lacrimal shares a broad articulation with the **antorbital (ao)**. The **supraorbital (so)** articulates with the posterior process of the antorbital bone. The circumorbital bones are not preserved on the left side of UO_B, giving the illusion of a large, deep suborbital cavity. Partly preserved and disarticulated circumorbital bones, including a right **dermosphenotic bone (dsph)** are preserved on the right side of UO_B and match the holotype in shape (Fig 3).

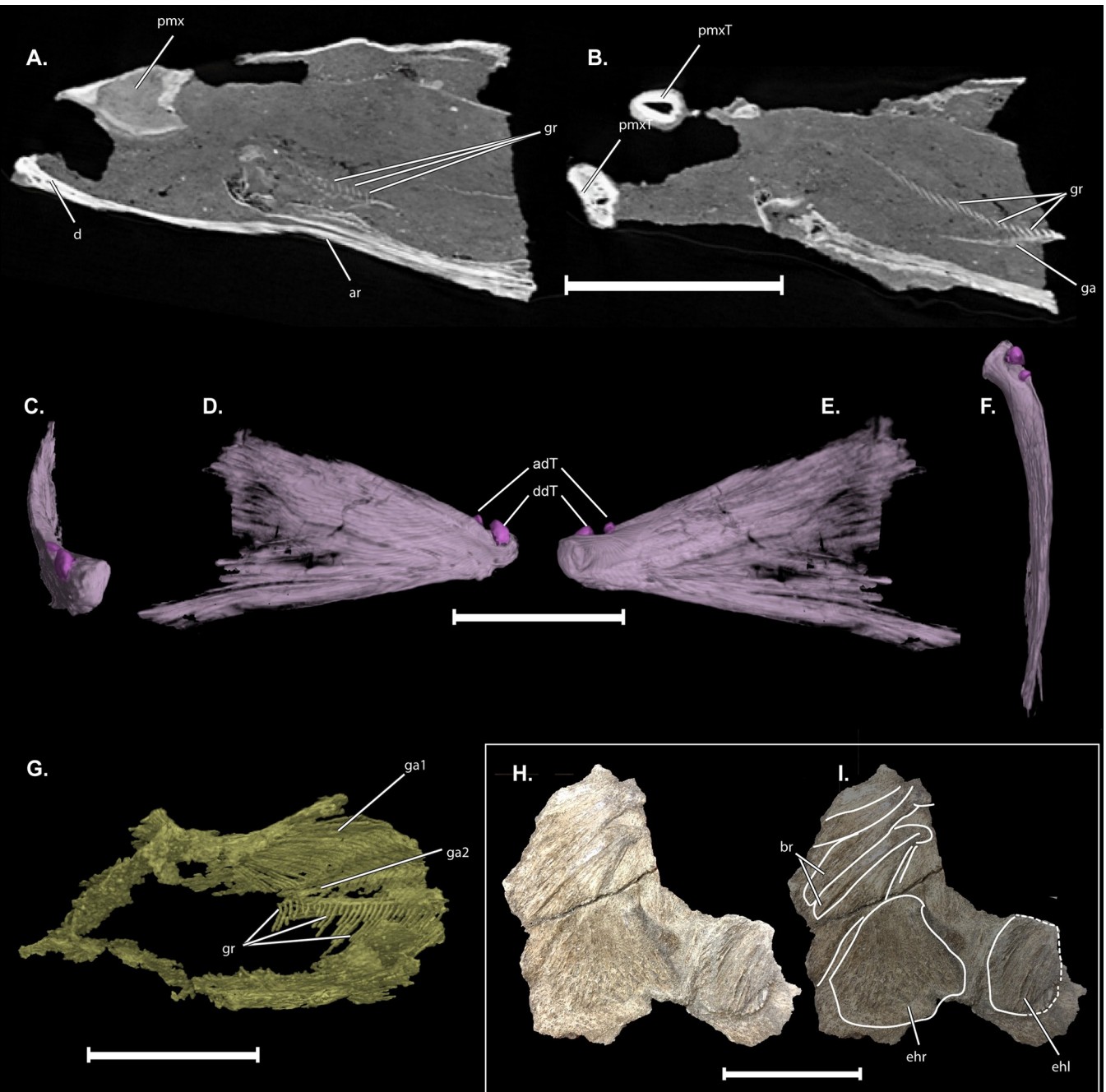

**Fig 6. Pharyngeal skeleton of †*Oncorhynchus rastrosus*, UO F-55101, specimen A.** (A) Proximal parasagittal cross section through dentary and hypertrophied premaxilla; (B) Distal parasagittal cross section through dentary and premaxillary spike; (C-F) right dentary in (C) anterior, (D) lateral, (E) lingual, and (F) dorsal views;(G) CT reconstruction of branchial basket; (H) dorsal view of posteroventral hyals; (I) line drawing of posteroventral hyals. A-C, anterior to left of page; D, anterior to top of page. Full width of page, portrait.

The **opercle (op)** is displaced from the opercular series in all specimens. It is highly ornamented externally (Fig 7) and somewhat rough internally. Most of the **preopercles (pop)** are intact on the left and right sides of UO_B (Fig 3). The **ventral limbs (popv)** of the left and right preopercles (horizontal limb [6] (Cavender & Miller, 1972); lower limb of [7] (Stearley &

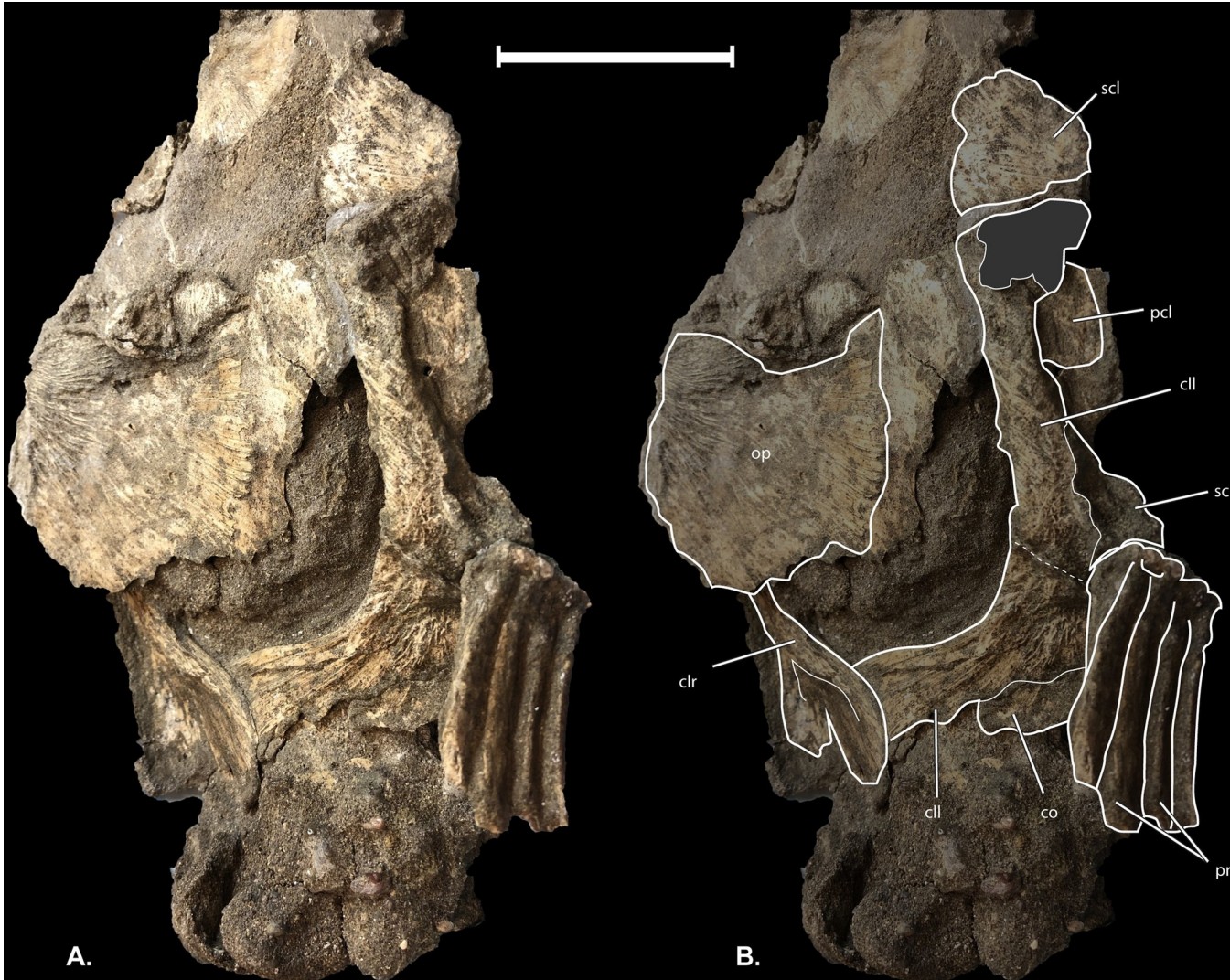

**Fig 7. †*Oncorhynchus rastrosus* UO F-55101, specimen C.** Pectoral skeleton and opercular series in left lateroventral view. Anterior to top of page. Scale bar = 5 cm. Full width of page, portrait.

Smith, Phylogeny of the Pacific Trouts and Salmons (*Oncorhynchus*) and Genera of the Family Salmonidae, 1993)) are slightly more downturned and shorter than the preopercles of the holotype (Fig 1A). Evidence of the **interopercle (iop)** is present on the right side of UO_B (Fig 3B). The **dorsal limb of the preopercle (popd)** does not contact the posterior circumorbital bones.

In the CT scan of UO_A (Fig 5A–5D), the median, diamond-shaped **vomer (v)** can be discerned along the anteroventral surface of the neurocranium. It is relatively broader at the head of the vomer than the head of the vomer in the holotype. No teeth are present on the vomer in the CT scan of UO_A. In cross-sectional scans, a poorly ossified, yet keel on the dorsal surface of the vomer is present, unlike the holotype. The vomer is preserved *in situ* articulating with the left-anteriormost portion of the **parasphenoid (ps)** on the primary block of OU_A. The right-anterior and posterior portion of the parasphenoid and left prootic are preserved in a secondary fragment of UO_A that was also CT scanned (Fig 5E).

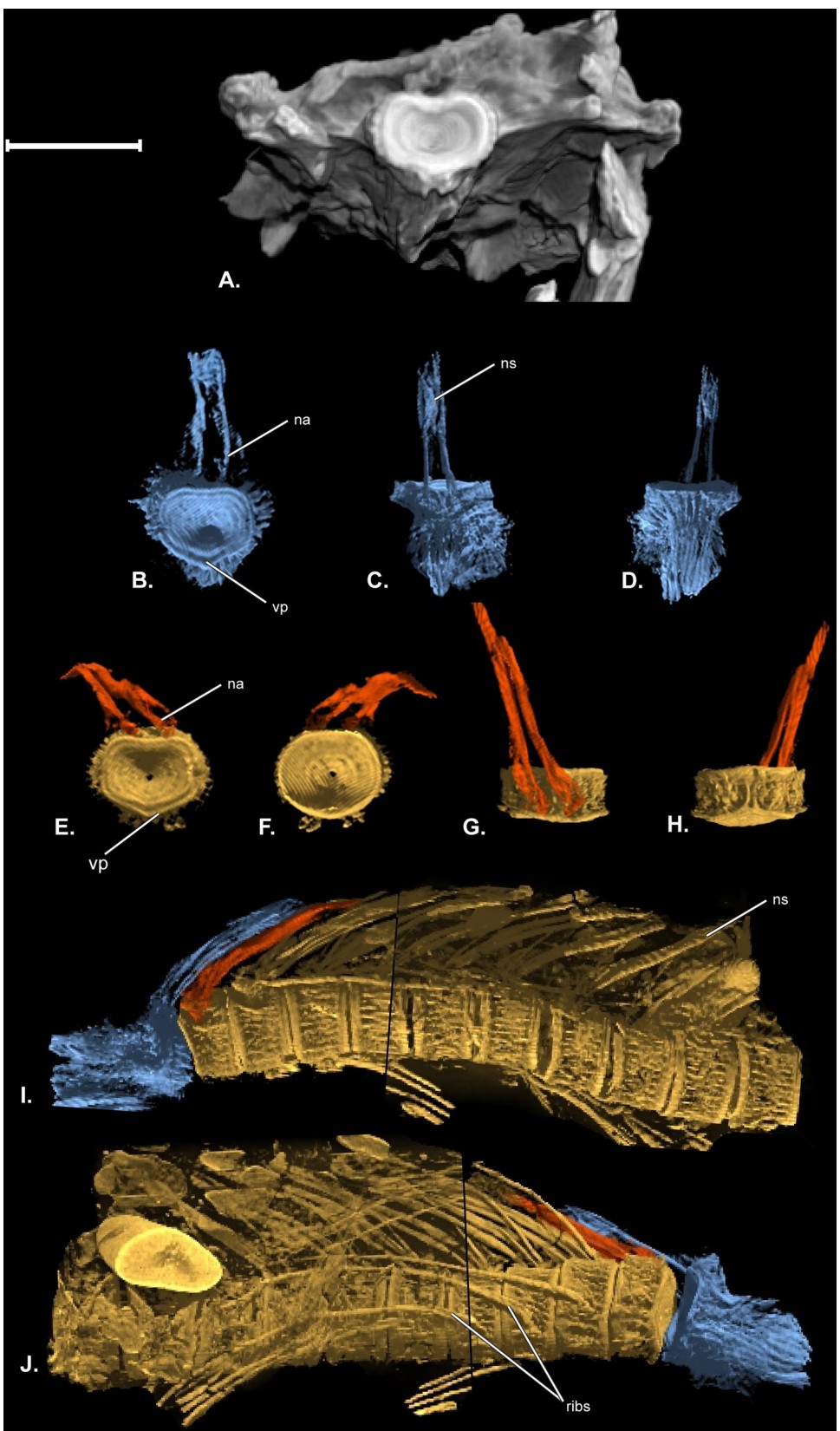

**Fig 8. CT renderings of anterior axial skeleton of †*Oncorhynchus rastrosus*.** (A) Posterior view of cranium of holotype UO F-26799; (B-J) UO F-55101, specimen C. (B) Posterior view of proatlas; (C) Dorsal view of proatlas; (D) Ventral view of proatlas; (E) Anterior view of first free centrum; (F) Posterior view of first free centrum; (G) Dorsal view of first free centrum; (H) Ventral view of first free centrum; (I) Left lateral view of cranio-vertebral articulation and anterior abdominal axial skeleton; (J) Right lateral view of cranio-vertebral articulation and anterior abdominal axial skeleton. Full page, portrait.

**Jaw and suspensorium.** Each **premaxilla (pmx)** bears an enormous **conical tooth (pmxT)** tipped with dark pigmented enamel (Fig 1B). While the premaxillae of the holotype are disarticulated, the paired premaxillae are articulated in UO_A and UO_B (Figs 1–6). Both premaxillae in UO_A bear intact teeth, while only the right premaxilla in UO_B preserves the tooth. A large socket on the left premaxilla of UO_B (Fig 3A and 3B) marks the original placement of the now missing dentition. These premaxillae have been described as ". . .constructed of the densest, heaviest bone possible, at great expense to construct and to carry around on the snout" (*pers. comm.* J. Smith to R. Troll, 16 April 2021). However, in cross sectional imaging, the bone appears not to be a solid mass (Fig 6A). Digital cross sections reveal a premaxilla comprising a dense cortical surface and a cancellous center, which renders as radiolucent in scans (Fig 6A).

The prominent premaxillary teeth of UO_A and U_B are directed laterally. The large conical premaxillary teeth are somewhat blunt in comparison to marine specimens of the same species. Each tooth contains a large pulp cavity (Fig 6B). Significantly smaller and sharper

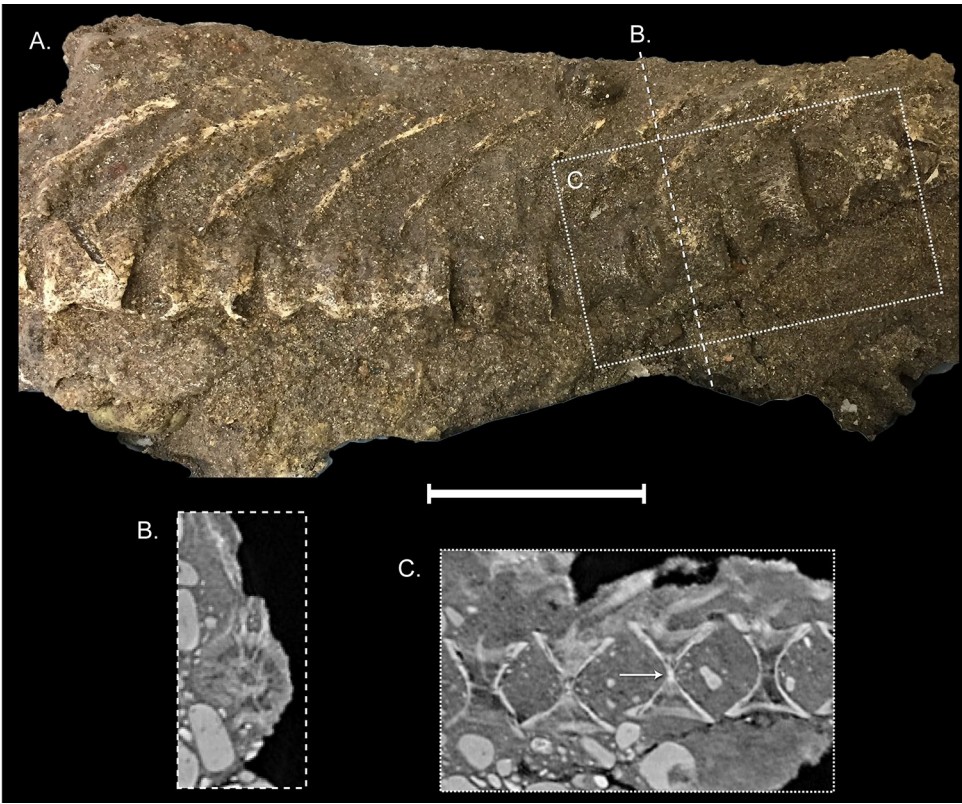

**Fig 9. Posterior abdominal series of †*Oncorhynchus rastrosus*, UO F-55101, specimen D.** (A) Left lateral view of articulated vertebrae; (B) digital axial cross section from dashed line, B; (C) digital median-sagittal cross section in box, C. full width of page, portrait.

**accessory teeth (aT)** are preserved posterior to the massive premaxillary tooth of UO_A (Figs 5A, 6C and 6D) and revealed in the digital section, within the matrix. The accessory teeth have a pulp cavity as well. It is unknown if these accessory teeth were preserved in UO_B. It should be noted that the accessory teeth are significantly proportionally smaller than the premaxillary teeth of marine †*O. rastrosus* described by Sankey et al. [8] (Sankey, et al., 2016) and appear to differ in tooth structure.

In the CT rendering of UO_A, with the laterally directed conical tooth, a premaxillary flange (**pmxf**; palatal process of Vladykov [28]) projects medially to presumably articulate with a median ethmoid cartilage (not preserved). A more delicate, thin, plate-like winged process (**pmxw**) equivalent to the "hinder wing" of Vladykov [28] (Vladykov, 1962)) is present on the true dorsal surface of the premaxilla (Fig 6D and 6E). That wing comes into close contact with the **anterior process of the maxilla (mxap)** (Fig 5B–5D). The anterior process of the **maxilla (mx)** projects medially at an approximately 45˚ angle from the **posterior process of the maxilla (mxpp)** in the holotype and in the **left maxilla (mxl)** of OU_A. There was distortion on the right side of the skull of OU_A, and the **right maxilla (mxr)** has a greater angle between the anterior and posterior processes. Teeth are absent from the maxilla, which is dense and not trabeculated in cross section. The **supramaxillae (smx)** in OU_A and OU_B have the same shape as in the holotype, however, the supramaxilla is proportionally smaller relative to the maxilla in OU_A and approximately the same proportions in OU_B.

Sagittal sections of the dentary of UO_A and UWBM-50816, of †*O. rastrosus* show a consistent, gracile texture across the entire edentulous and tooth bed zones (Fig 6A; see also S1 Video). In μCT scan sagittal sections of †*O. rastrosus* UWBM-71908a and 71908b [13: Figs 4.28 and 4.30], there is a honeycomb-like network of hollows in the shaft of the dentary. The distal tooth bed is bulky and full of many small, densely packed channels. It is slightly distinguished from the tooth bed itself by a rough, nodular texture present anterior to the tooth bed. The dentaries are nearly triangular and dorsoventrally deeper than other species of *Oncorhynchus*. Dentaries appear trabeculated in CT scans (Fig 6B) and possess a rough outer surface. While lacking evidence of feeding teeth, each dentary of UO_A and UO_B bears at least one minute, laterally directed tooth (**dT**) at its mesial end near the symphysis (Figs 2 and 3). UO_A also preserves an accessory distal tooth as noted in the holotype [6] (Cavender & Miller, 1972).

Left and right **articular bones (ar)** are present in UO_A and UO_B although the quadrate facet of the right articular bone is not preserved on UO_A (Fig 2A and 2B). Each articular is triangular with a tall ascending ramus that is roughly the same length as the ventral ramus. There is no notch between the two rami, and thus the anterior margin of the articular is roughly perpendicular to its ventral margin. A short projection from the ascending ramus of the articular and a long prominent **retroarticular process (rarp)** contribute to a deep quadrate facet (qf).

The left **quadrate (q)** is partially preserved in UO_A and both are preserved in UO_B. In all preserved quadrates, the **articular process (apq)** is a massive condyle bounded by a deep quadrate facet (Figs 2A, 2B, 3, and 5). The broad fan-like dorsal process of the quadrate is directed somewhat medially towards the **endopterygoid (end)**. On the right side of UO_B (Fig 3B), there is a prominent **metapterygoid (mpt)** superficial to the endopterygoid and slightly dorsal to the quadrate, similar to what is preserved on the left side of the holotype. There is also a long slender **posterior process of the quadrate (ppq)** that extends the length of the **symplectic bone (sy)**. This process is longer than in the holotype (Figs 1A, 2, and 3). A slender ventral process of the left **ectopterygoid (ect)** is preserved in UO_A (Fig 2A and 2B). The right ectopterygoid is nearly complete in UO_B and does not differ from the holotype (Fig 3A and 3B).

**Branchial skeleton.** Mechanical preparation exposed the branchial skeleton of the †*Oncorhynchus rastrosus* holotype in dorsal view. The CT scan of UO_A allows for digital preparation of the oral cavity, which was filled with coarse sand and gravel, as was that of the

holotype. The sand and gravel are less dense than the bone, and thus many of many elements of the branchial skeleton were identifiable. The *in-situ* t-shaped **basihyal (bh)**, has a blunt anterior end, and lacks teeth (Fig 5A–5C). Posterior to the basihyal are paired **hypohyals (hh)**. The anterior end of each hypohyal is hooked medially, though are otherwise bulbous. Hypohyals lie proximal to broad, rectangular **ceratohyals (ch)**, though they are not in direct contact. The ceratohyals are somewhat ornate, preserving an x-shape pattern on the lateral surfaces. Between the hypohyals and ceratohyals is a long median **urohyal (uh)**. The urohyal is blunt on its anterior face. It tapers posterior to the face and then widens posteriorly. It is dorsoventrally deepest and slightly forked laterally at its posterior limit. The **gill arches (ga)** follow the pattern seen in the holotype. Two ventral limbs of the gill arch are identifiable in the CT scans and model of OU_A; the lateralmost is noted as a portion of **gill arch 1 (ga1)** and the ventralmost is noted as **gill arch 2 (ga2)**. There are hundreds of **gill rakers (gr)** preserved across all arches. Roughly 30 rakers are identifiable on the partially preserved gill arch 1, which appear more elongate and gracile than the 30+ stout rakers on each gill arch 2 (Fig 6C). An **epihyal (eh)** is preserved in the post-cranial specimen, UO_C, with overlapping **branchiostegal rays (br)**.

**Pectoral skeleton.**    The holotype of †*Oncorhynchus rastrosus* preserved only a fragment of the **cleithrum (cl)** and **posttemporal (pt)** bones. Specimen UO_C adds much more information about the pectoral skeleton (Fig 7). The paired cleithra are broad and flat. The cleithra are slightly anteriorly shifted exposing the **coracoid (co)** bones, left **scapula (sc)**, and **left postcleithrum (pcl)**. An isolated, robust left **supracleithrum (scl)** is also preserved. Medially, the paired coracoid bones are wide and overlap one another. The lateral portion of the coracoid articulates with a rectangular scapula. A series of four **proximal radials (prr)** are tightly associated with each other and situated adjacent to the scapula. No distal fin rays are present on the fin of UO_C, however some distal fin rays were found among the paratype specimens. Distal fin rays are narrower than the proximal fin rays and divided into two **hemitrichia (hem)**. Proximal fin rays and distal fin ray hemitrichia are not segmented. The proximal ends of dorsal hemitrichia have overlapping processes.

**Proatlas, vertebrae, and median fins.**    The posterior view of the holotype shows the proatlas which is dorsoventrally flat and laterally broad, with a median ventral V-shaped peak (Fig 8A). Newly recovered vertebrae are all abdominal and these are dorsoventrally taller than they are anteroposteriorly long (Figs 8 and 9). All centra recovered are anteroposteriorly longer than the caudal centrum of the holotype photographed in Fig 11 of Cavender and Miller [6] (Cavender & Miller, 1972). Vertebral centra, including the proatlas, are ornamented by many fine, elevated, longitudinal ridges (Fig 8B–8J). In posterior view (Fig 8B), the proatlas of UO_C is dorsoventrally compressed and there is a short **ventral V-shaped peak (vp),** similar to the holotype. There is a tall **neural arch (na)** and a median **neural spine (ns)** associated with the proatlas (Fig 8B and 8H). Grooves and ridges of the articular face of the proatlas match those on the adjacent anterior face of the first free centrum (Fig 8E).

The first free vertebral centrum is anteroposteriorly shorter than remaining abdominal vertebrae, which get progressively longer until the fifth centrum after which adjacent centra are approximately equal in size (Fig 8I and 8J). In anterior view, the neural arches of the anteriormost abdominal vertebrae are tall and neural spines are not bifurcate (Fig 8F). In lateral view, the anterior neural spines are closely associated, laterally compressed, and posteriorly directed (Fig 8I and 8J). More posteriorly, the neural spines are singular and rod-like (Fig 9A). Neural arches are not fused to the centrum. Ribs are present in articulation with the second, third, and fourth free vertebral centra.

The posterior abdominal series is well developed with some diagenetic deformation. Each vertebra is roughly cylindrical with the centrum made up of two hollow cones fused at their apices. In the axial cross section, cones exhibit a highly trabeculated mineralization pattern

(Fig 9B). The two cones communicate with each other through a small opening that is secondarily ossified in these fossils (Fig 9C).

## Phylogenetic results

The parsimony analysis recovered 60 most parsimonious trees at 257 steps, a consistency index (CI) = 0.504, homoplasy index (HI) = 0.496, retention index (RI) = 0.841, and rescaled consistency index (RC) = 0.424. The consensus tree appears in Fig 10A. The relationships resemble those of Stearley and Smith [7], but now include several extinct filter feeding taxa. †*Oncorhynchus rastrosus* is sister to †*O. rastellus*, united by a deep dentary (character 64) and a bulbous premaxillary bone (not included in the character matrix). Together, these taxa are

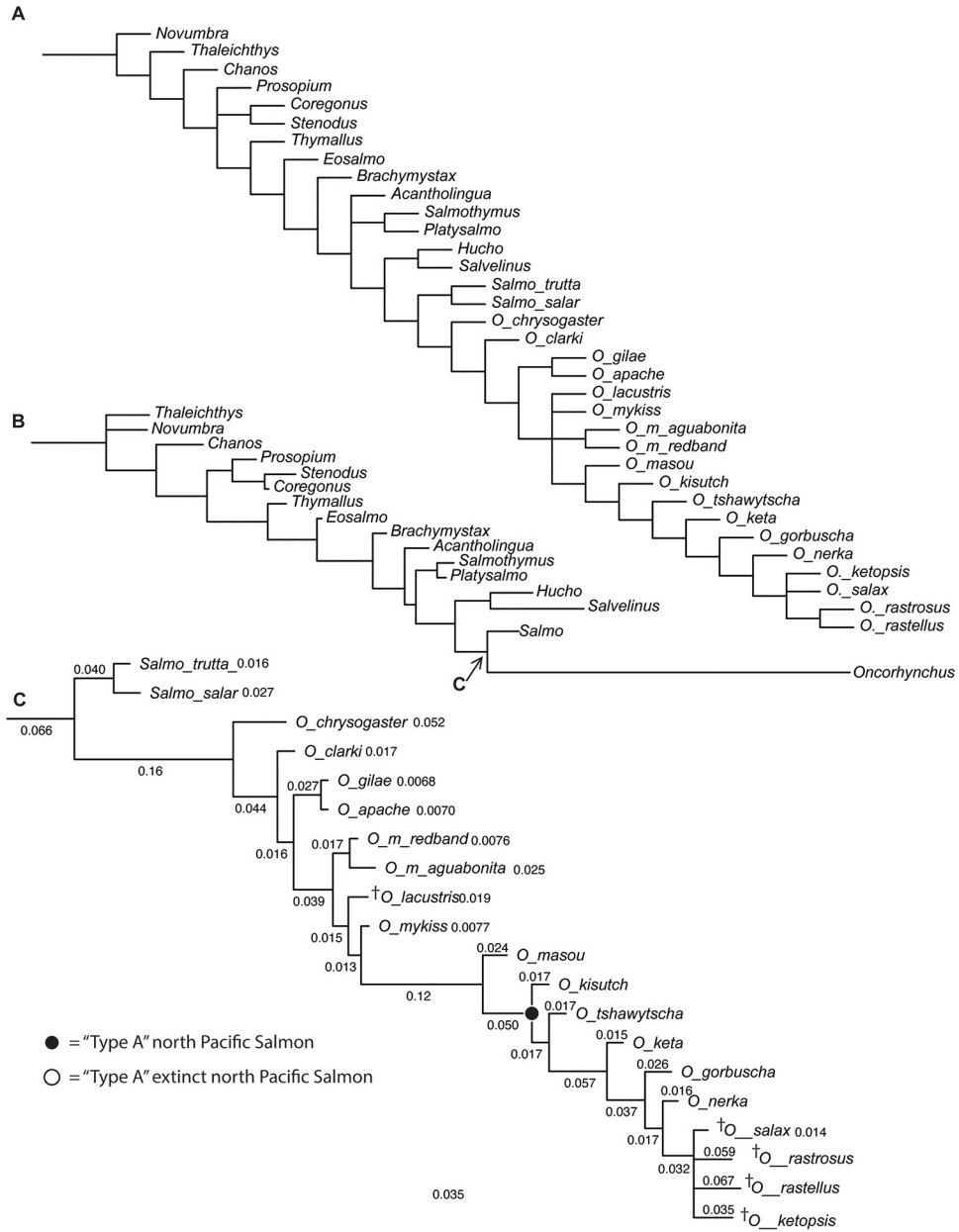

**Fig 10. Phylogenetic hypotheses of Salmonidae.** (A) parsimony hypothesis of all OTUs; (B) Bayesian hypothesis of salmonid genera; (C) expanded relationships of extant and extinct species of *Oncorhynchus*. Full page, portrait.

highly nested within a clade of extinct filter feeding Pacific salmon, separated from *O. nerka* due to the presence of dark enameloid on the inside of the teeth (character 70). *Oncorhynchus nerka* and the fossil clade are united by a preopercle ventral limb reduced to half the length of dorsal limb (character 81). Together they are sister taxon to *O. gorbuscha* and united by frontals contacting epiotics and a dorsal hump in breeding males, developed to an extreme (characters 17, 106, and 107). Finally, those taxa are sister to *O. keta*, united by five unambiguous transformations (characters 8, 36, 51, 59, and 87). Thus, the filter feeding fossils are derived members of the "Type-A" Pacific salmons *sensu* Hikita [29] (Hikita, 1962).

Bayesian analysis (Fig 10B and 10C) also recovers a derived clade of "Type-A" clade of extant Pacific salmons, where Coho, Chinook, Chum, Pink, and Sockeye salmon form successive sister taxa (*O. kitutch* + (*O. tshawytscha* + (*O. keta* + (*O. gorbuscha* + (*O. nerka*))))). It further nests the newly included extinct taxa within the "Type-A" salmon in a clade that is sister to *O. nerka* (Fig 10C). Within the Mio-Pliocene clade, the planktivorous species †*Oncorhynchus rastrosus* and †*O. rastellus* are sister taxa, supporting the hypothesis proposed by Stearley and Smith [9] (Stearley & Smith, Salmonid Fishes From Mio-Pliocene Lake Sediments in The Western Snake River Plain and The Great Basin. Fishes of the Mio-Pliocene Western Snake River Plain and Vicinity: Museum of Zoology, University of Michigan, 2016). The Miocene taxa, †*O. ketopsis* and †*O. salax* form successive sister taxa to the planktivorous species (†*O. salax* + †*O. ketopsis* + †*O. rastrosus* + †*O. rastellus*) whose relationships lack resolution in the Bayesian tree.

## Discussion

### Interrelationships of Pacific salmon

Molecular hypotheses of salmonine interrelationships vary in the number of taxa and genes sampled [30–33]. While fossils were used to calibrate these molecular hypotheses, total evidence analyses were not conducted and thus, we consider the relationship of †*O. rastrosus* and other Mio-Pliocene salmonines in the context of morphology. For that the best data come from Stearley and Smith [7], who examined the relationship of extant and extinct Salmonidae with special reference to *Oncorhynchus*. Our parsimony analysis recovers a relationship (Fig 10A) among members of *Oncorhynchus* that is consistent with their consensus hypotheses [7: Figs 10 and 12].

Our Bayesian analysis recovers a polytomy among all extinct Mio-Pliocene taxa (Fig 10A and 10C). Bayesian and parsimony analyses recover a derived clade, "Type-A" clade, of extant Pacific salmons. The "Type-A" salmon are non-monophyletic in molecular hypotheses. Despite extensive taxon and gene sampling of *Oncorhynchus*, an unequivocal portrait of the relationships of species of *Oncorhynchus* appears difficult to obtain [30] (Crête-Lafrenière, Weir, & Bernatchez, 2012). Among the hypotheses, one relationship remains consistent, (*O. nerka* + (*O. keta* + *O. gorbuscha*)). Our morphological results place the newly included extinct Mio-Pliocene taxa nested within "Type-A" salmon in a clade that is sister to *O. nerka* (Fig 10C). Within the Mio-Pliocene clade, the planktivorous species †*Oncorhynchus rastrosus* and †*O. rastellus* are sister taxa, supporting the hypothesis proposed by Stearley and Smith [9].

The Bayesian analysis also recovers †*Oncorhynchus lacustris* as stem to *O. mykiss* + "Type A" Pacific salmon, rather than unresolved with respect to sub-species of *O. mykiss*. This analysis is relevant considering recent molecular hypothesis (Fig 11) of *Oncorhynchus* relationships, focusing on Mexican taxa [32] (Colín, Del Río-Portilla, Lafarga-De la Cruz, Ingle-De la Mora, & García-De León, 2023). In that study, *O. chrysogaster* (Mexican Golden Trout) resolves as sister taxon to *O. mykiss nelsoni*, rather than to *O. gilae* as originally predicted. Previous work indicated that the *O. mykiss* (Rainbow Trout), California Golden Trout, and Redband Trout

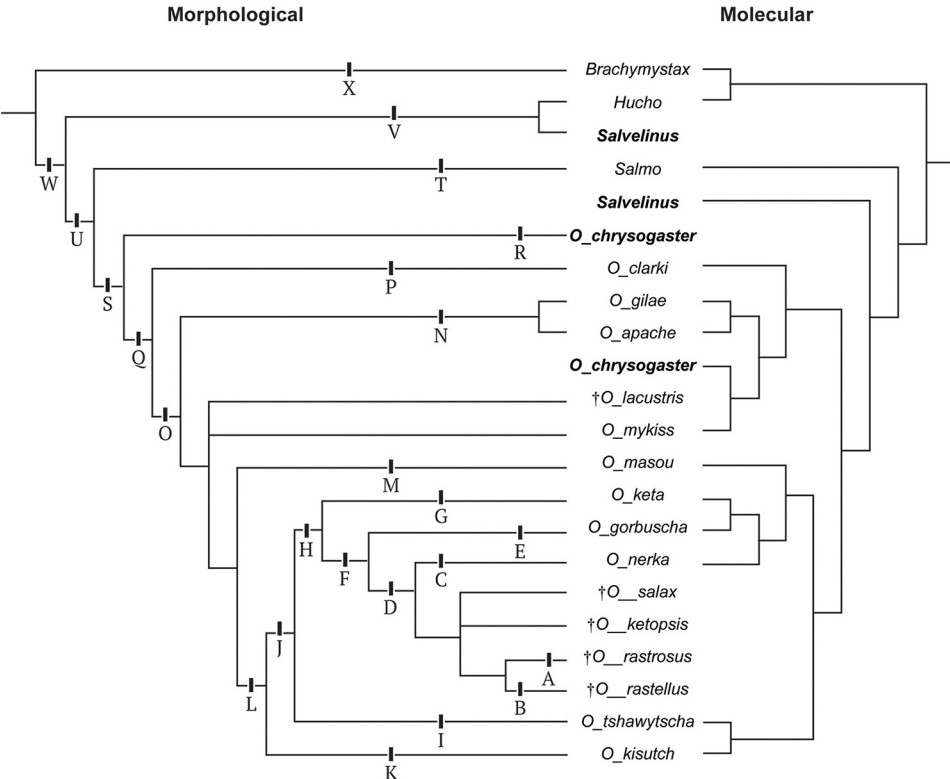

**Fig 11. Character optimization on the present study's morphological hypothesis, juxtaposed with the prevailing molecular hypothesis.** Molecular tree modified from Crête-Lafrenière et al. (2012), Horreo (2017), and Colín et al. (2023). Bold-italicized taxa appear twice on the figure and once on each tree. Unambiguous character transformations for labeled branches where bold text means CI = 1.0: A = 49:0>1; B = 67:1>0, 91:1>0; C = **70:0>1**; D = 81:2>1; E = 72:2>1, **107:0>1**; F = **17:0>1**, 101:0>1, **106:0>1**; G = 98:0>1; H = **8:0>1**, 36:1>0, 51:2>1, **59:0>1**, 87:0>1; I = **28:0>1**; J = **7:1>2**; K = 22:0>1; L = **1:0>1**, 93:1>2, 105:1>2, 114:1>0; M = 98:0>1, **118:0>1**; N = 76:0>1, **113:0>1**. Full width of page, portrait.

are a natural group of Pacific salmon while Cutthroat Trout and Mexican Golden Trout are outside the Pacific salmon rainbow trout group [7]. Given that the Mexican Golden Trout does not cluster with the Gila or Apache Trout [32], this resolution could support the hypotheses that Redband Trout origins "...lay near the Gulf of California..." (Behnke in [9]).

## Feeding ecology

†*Oncorhynchus rastrosus*, estimated between 2.4 m– 2.7 m [6, 9], was a giant anadromous fish that would have spent a substantial portion of time feeding in ocean waters. Marine suspension feeders are the largest vertebrates alive today and have been among the largest recovered in the fossil record [34] (Thewissen & Waugh, 2023). Historically, suspension feeding niches were dominated by cartilaginous lineages (sharks and rays) and even marine mammals (baleen whales). Among bony fishes, there are only a few examples of large suspension feeders, including the North American freshwater paddlefish, *Polyodon*, and the Mesozoic (e.g., †*Asthenocormus*, †*Leedsichthys*, and †*Rhinconichthys*) [35] (Friedman, et al., 2010). †*Oncorhynchus rastrosus*, like these large-bodied marine suspension feeders, possessed mostly edentulous dentaries, elongate maxillae, enlarged gill rakers, and indeed, more gill rakers than any other salmon. In ensemble, those features suggest that it also consumed plankton. Unlike those large suspension feeders, the dentary is relatively short. Therefore, it seems unlikely †*O. rastrosus*

would have been able to achieve a comparable gape. The mouth in †*O. rastrosus* also is proportionally smaller relative to the head length and estimated body length than in extant salmonids, which primarily occupy predatory niches. Extant salmon have non-protrusible mandibles as well as a modified tongue bite apparatus, which allows for a specific prey-capture and feeding strategy.

These differences in jaw structure from their extant relatives and from other large planktivores leads us to consider whether †*O. rastrosus* ingested their food using an alternative mechanism. For example, rather than strictly ram feeding as in other known large bodied, pelagic, planktivorous vertebrates, the massive opercular series and loose attachment of the premaxilla may have allowed for some degree of suction feeding on clouds of plankton. Given the unusual shape of the premaxillae, it is conceivable that there could have been at least some rotation involved in the upper jaw of †*O. rastrosus*, or maybe even slight protrusion of the premaxillae to expand the buccal chamber and create negative pressure. We suspect that they achieved premaxillary rotation or flaring because the premaxillae of fossil specimens were routinely found dissociated, suggesting a loose ligamentous connection to the rest of the skull.

The complex branchial skeleton and strong triangular lower jaw that differs substantially in shape from living salmonids could also have been used to scoop sediment. Such behavior is known in several taxa. For example, the twostripe goby, *Valenciennea helsdingenii*, scoops mouthfuls of surface sand and discards most of it through their gill openings [36] (Clark, Stoll, Alburn, & Petzold, 2000); the detritivorous Curimatidae of South America have complex fleshy projections in the mouth and epibranchial organs that assist the winnowing process [37,38] (Vari, 1989; Melo, et al., 2018); and Grey Whales (*Eschrichtius robustus)* are known to scoop mud filled with benthic organisms [39] (Pivorunas, 1979). While these organisms represent extremes of body size and niche, they demonstrate that filtering is not only for siphoning food, but also for discarding inorganics. Thus, benthic sediment feeding represents another possible feeding ecology for †*Oncorhynchus rastrosus*.

## Sexual dimorphism

Salmons, particularly males, are known for exceptional transformations while migrating to spawn [28,40] (Vladykov, 1962; Tchernavin, 1938). Development of the kype is a synapomorphy of the Eusalmoninae, and Pacific and Atlantic salmon have evolved their own variations of kype metamorphosis. The transformations are most pronounced among species of the Pacific genus *Oncorhynchus*, whose generic name refers to the remarkable "hooked noses" of breeding males.

The kypes of *Oncorhynchus* and *Salmo* differ internally as well as externally. Pacific salmon kypes include a smaller array of shorter, squatter skeletal needles combined with several robust, resource intensive breeding teeth. Atlantic salmon kypes on the other hand have long, delicate skeletal needles, which would be a hindrance, costly, and easily damaged if present in *Oncorhynchus* [13] (Prescott., 2014). These changes are even preserved in the fossil record of several salmonine taxa, including †*O nerka*. For instance, in sagittal section of (UWBM-87121 from Skokomish Valley, Washington), the tooth bed of the dentary has a spongiose structure that transitions into a more compact, linearly arranged bone posteriorly.

The more elaborate changes observed in extant Pacific salmon, likely reflect a marked advantage for the males during their semelparous reproduction [41–44]. Because Atlantic salmon are iteroparous, with a proportion of mating individuals surviving the spawn run to mate in a following year, they are not able to commit as many resources to kype development and may accordingly experience less extreme selection for dimorphism [13].

In addition to the kype, three additional elements, the vomer, rostro-dermethmoid-supraethmoid, and premaxillae are remodeled as breeding approaches. Among extant species of *Oncorhynchus*, the length of the head of the vomer correlates with the length of the rostrum and width of the head. In extant breeding males, the vomer is more elongate than in breeding females of the same species [28]. Immature males and females of extant *Oncorhynchus* possess a rostro-dermethmoid-supraethmoid with similar dimensions within species. Mature males, however, experience a pronounced growth in their ethmoid region and possess a much larger rds bone than do females. This is particularly noticeable in *O. keta*, *O. masou*, and *O. kisutch* and "...no bone in the skull of *Oncorhynchus* is more strongly affected by breeding growth than the premaxillary. In males it practically doubles in length from the clean [*oceanic*] to the breeding stage [28: p. 102]."

Thus, we expected †*Oncorhynchus rastrosus* to show similar dimorphism of the vomer, rds, dentary, and premaxilla. Our examinations of these elements in multiple specimens uncovered evidence of dimorphism in the dentary, vomer, and rds, but not the premaxillae nor premaxillary teeth. †*Oncorhynchus rastrosus*, unlike other species of *Oncorhynchus*, does not exhibit an extreme kype. We therefore examined high resolution µCT scans of †*O. rastrosus* for evidence of kype-associated bone tissue or other dimorphism, even in the absence of an extreme external kype. Sagittal sections of the dentary of UO_A and UWBM-50816 show a consistent, gracile texture across the entire edentulous and tooth bed zones (Fig 6A; see also S1 Video). In comparison to sagittal sections of UWBM-71908a and 71908b, the distal dentary, tooth bed, and edentulous shaft are not uniform. The honeycomb-like network of hollows in the shaft of the dentary and nodular texture present anterior to the tooth bed is akin to that observed in *Salvelinus* specimens and, in sagittal section, would appear to have a similar density gradient. While this region appears to be continuous with the bone, there is some variation from the rest of the dentary in terms of its internal organization.

The tooth morphology on the dentary varies in these specimens as well. There are two teeth in UO_A and UWBM-50816, the mesial most tooth slightly larger than the distal tooth. UWBM-71908a and 71908b have a single, large tooth with a slightly bulbous tooth base. Given the uniform density in UO_A compared to the variable density noted in the thesis work of ZMP and dentition described herein, we propose these characters as evidence supporting the presence of sexual dimorphism in the dentary of †*Oncorhynchus rastrosus*. Furthermore, we hypothesize that the UO_A specimen is female, and the holotype is a male (Table 1).

Metamorphic change also occurs in the vomer of breeding males of extant *Oncorhynchus* species [28] (Vladykov, 1962), contributing to sexual dimorphism. Specifically, male vomers become relatively longer during the drastic rostrum elongation in breeding males. Vladykov [28] noted that in *O. keta* there is a prominent keel, reminiscent of a sailboat. We clarify here that this "keel" is dorsal, rather than ventral as would describe a stabilizing hull structure in sailboats. The vomer in †*Oncorhynchus rastrosus* is an elongated diamond with no pronounced constriction in the holotype and UO_A. The vomer of UO_A is broader anteriorly than it is posteriorly. In the holotype of †*O. rastrosus*, the vomer is thickened anteriorly with a spoon-like bevel to the lateral margins of the vomeral head, reminiscent of paired, parallel keels. In UO_A, however, the bevel is less pronounced. In lateral view, the vomer of UO_A resembles the breeding female morphology of *O. gorbuscha*, *O. tshawytscha*, and *O. nerka*, which have a moderate arch on the dorsal surface. No teeth are found in the holotype of †*O. rastrosus* or in UO_A, which are generally found in breeders of extant taxa. The differences seen between the vomer bones of the holotype and UO_A is a second line of evidence for sexual dimorphism in †*O. rastrosus*, with the holotype representing the male morphology and UO_A representing the female form.

**Table 1. Sexually dimorphic character suites by specimen.**

|  | UO_A | UWBM50816 | UO F-26799 | UO_B | UO F-3335 | UWBM71908a | UWBM71908b |
|---|---|---|---|---|---|---|---|
| **1. Kype A** | 0 | 0 | ? | ? | 1 | 1 | 1 |
| **2. Kype B** | 0 | 0 | 1 | 1 | 1 | 1 | 1 |
| **3. Vomer A** | 1 | n/a | 0 | ? | ?* | n/a | n/a |
| **4. Vomer B** | 0 | n/a | 1 | ? | ?* | n/a | n/a |
| **5. RDS** | 0 | n/a | 1 | 1 | ?* | n/a | n/a |
| **6. Premax** | 1 | n/a | 1 | 1 | 1 | n/a | n/a |
| **Sex** | F | F | M | M | M | M | M |

Characters and states: 1. Kype A: (0) absent; (1), present; 2. Kype B, dentary teeth: (0) multiple small teeth, (1) single tooth on bulbous base; 3. Vomer A, prominent dorsal sail: (0) absent, (1) present; 4. Vomer B, length: (0) laterally broad, (1) elongate; 5. RDS shape: (0) anteriorly blunt and wide, (1) anteriorly pointed and narrow; 6. Premaxilla: (0) non-bulbous, (1) bulbous; Predicted sex: (F) female, (M) male.

*specimen is missing these elements according to Cavender and Miller (1972)

? = missing

n/a = not applicable because specimen was only dentary

A third skeletal element also leads us to suspect that UO_A is a female while the holotype and UO_B are males. The rostro-dermethmoid-supraethmoid is the dermal bone medial to the paired premaxilla. A blunt rostral margin of the rds occurs in female (immature and mature) or immature male specimens of extant Pacific Salmon, while in breeding males, the rostral portion of the rds is more elongate and/or pointed [28] (Vladykov, 1962). The CT scan of UO_A confirms an anteriorly blunt and posteriorly notched rds, consistent with female morphology. The anterior rostral portion is shorter and broader than in UO_B and in the holotype, while posterior wings are relatively shorter in UO_A.

In contrast, the premaxilla of †*Oncorhynchus rastrosus* does not demonstrate sexual dimorphism, though it does undergo ontogenetic morphological change in breeding males and females to an equal extent. In all specimens excavated from upstream, freshwater localities for this study and previously published studies, the premaxillae are bulbous with prominent, slightly worn dentition [6,8,9] (Cavender & Miller, 1972; Sankey, et al., 2016; Stearley & Smith, Salmonid Fishes From Mio-Pliocene Lake Sediments in The Western Snake River Plain and The Great Basin. Fishes of the Mio-Pliocene Western Snake River Plain and Vicinity: Museum of Zoology, University of Michigan, 2016). The premaxilla of specimens recovered from coastal marine environments is much less bulbous, close in size to the tooth it bears [8] (Sankey, et al., 2016).

## Spikes, not fangs

Though the saber-like premaxillary teeth of †*Oncorhynchus rastrosus* have brought substantial fame to the species, prior reconstructions of that tooth's orientation apparently erred. The originally described premaxillae were recovered as isolated elements and, though they were associated with skull material, they were never preserved *in situ* [6,8] (Cavender & Miller, 1972; Sankey, et al., 2016). Because these premaxillae lacked evidence of the processes and facets that could reveal how they articulate with the ethmoid cartilages and maxillae, it is understandable that the premaxillae were reconstructed with a ventrally directed tooth [6] (Cavender & Miller, 1972), as otherwise observed among extant salmonids [28] (Vladykov, 1962). In the original configuration (Fig 1A), the teeth resembled the canine teeth of the extinct felids in *Smilodon* [45] (Lund, 1842) and the taxon was originally named "†*Smilodonichthys*" *rastrosus* [6] (Cavender & Miller, 1972). Even after placement within *Oncorhynchus*

[7] (Stearley & Smith, Phylogeny of the Pacific Trouts and Salmons (*Oncorhynchus*) and Genera of the Family Salmonidae, 1993), the taxon remained known as the Sabertoothed Salmon.

The newly recovered specimens bear *in situ* premaxillae with teeth resembling spikes, not fangs, and thus we colloquially rename the species the "Spike-toothed Salmon". The exceptional preservation of the premaxillae in UO_A and UO_B leave no doubt that the teeth of †*Oncorhynchus rastrosus* projected laterally from the skull, creating a morphology resembling the spiked tusks of muntjacs (*Muntiacus* spp.) [46] (Chapman, Chapman, & Coles, 1985) or warthogs (*Phacochoerus* spp.) [47] (Grubb & d'Huart, 2010). A key feature on the premaxillae that allows us to orient isolated premaxilla is the wing-like thin, bony-plate on the posterodorsal aspect, which was worn down, overlooked, or misinterpreted in isolated, previously described specimens [6,8,9] (Cavender & Miller, 1972; Sankey, et al., 2016; Stearley & Smith, Salmonid Fishes From Mio-Pliocene Lake Sediments in The Western Snake River Plain and The Great Basin. Fishes of the Mio-Pliocene Western Snake River Plain and Vicinity: Museum of Zoology, University of Michigan, 2016).

While overall size of the premaxilla and spike increased when migrating and maturing [8] (Sankey, et al., 2016), the relative position of the premaxilla during the oceanic phase remains uncertain. It is possible that there was some rotation from a more "traditional" position in marine spike-tooth salmon prior to moving instream. Unfortunately, among coastal marine specimens the premaxillary bone itself was too worn to identify a premaxillary wing or premaxillary flange and test the hypothesis of premaxillary rotation during maturation.

No other member of the salmoniform radiation, at any life stage, possesses teeth with such an unusual orientation. The unique occurrence of this morphology in the largest known salmon likely indicates that the teeth were used forcefully and were probably powered by the axial body musculature. In actinopterygian and chondrichthyan fishes, the epaxials and hypaxials can apply substantial lateral force [48] (Nauen & Lauder, 2002) and power both swimming and rapid mouth expansion during suction feeding [49,50]. The extensive body musculature in †*O. rastrosus* would have provided the impressive lateral forces needed to deploy the spike teeth in a deliberate way, and the laterally widened craniovertebral joint may have stabilized the cranium during lateral undulation. We consider here the possibilities that spikes were offensive or defensive weapons, aided competition, or functioned as tools.

**Offense/Defense.**   Among other aquatic vertebrates, enlarged teeth function most frequently as offensive weapons. For example, barracuda (*Sphyraena barracuda*) dismember prey larger than their gape [51] (Grubich, A.N., & Westneat, 2008), chondrichthyans cut, tear, puncture or crush [52] (Whitenack & Motta, 2010), Lancetfish (*Alepisaurus ferox*) slash with their hypertrophied teeth [53] (Romanov & Zamorov, 2002), dogtooth characins (*Hydrolycus* spp.) puncture their prey before swallowing it whole [54] (Goulding, 1980), and mesopelagic predators such as dragonfishes (*Chauliodus*) [55] (Greven, Walker, & Zanger, 2009) and fangtooths (*Anoplogaster*)[56] (Kierdorf, Kierdorf, Greven, & Clemen, 2022) cage their prey. Even groups of teleost not normally known for elaborate oral dentitions have evolved fanged representatives, such as the sabertoothed anchovy *Monosmilus chureloides* [57] (Capobianco, et al., 2020), a fossil predator from the Eocene in a group normally known for planktivory, or *Danionella dracula*, a miniature minnow with fanglike odontoid process on its jaw bones [58] (Britz, Conway, & Rüber, 2009). Though these disparate predatory taxa possess disparate tooth morphologies, all have teeth that project ventrally into the gape rather than laterally from the mouth.

Laterally or anteriorly directed teeth are comparatively rare among actinopterygian fishes. Most famously, Neotropical characiforms in the genus *Roeboides* use anteriorly directed premaxillary teeth to ram and de-scale other fishes [59,60] (Novakowski, Fugi, & Hahn, 2004; Kolmann, Huie, Evans, & Summers, 2018). Though a lepidophagous tooth function would be

remarkable, is not a parsimonious explanation for †*O. rastrosus* given the numerous gill rakers which suggest strongly that the animal was planktivorous [61] (Eiting & Smith, 2007). Marine platytroctids (*Holtbyrnia facilis*, *H. anomala*) provide another parallel by having up to four, forward-directed premaxillary tusks [62] (Matsui & Rosenblatt, 1987). The function of those tusks appears to be unknown, but the species are zooplanktivorous [63] (Novotny, 2018) and no evidence has linked their teeth to filter-feeding.

Many actinopterygians evolve spines or spikes as defensive structures, often as a passive means to confound gape limited predators [64,65] (Bosher, Newton, & Fine, 2006; Price, Friedman, & Wainwright, 2015). In †*Oncorhynchus rastrosus*, a passive defensive function can be ruled out because the teeth do not contribute meaningfully to the girth of this massive animal, and predators large enough to attack it (e.g. sharks, sea lions, or toothed whales) would likely use a dismemberment strategy. However, an active defensive function, in which the animal used the tusks to hit or ward off predators, is plausible–particularly given that males and females both possess the enlarged teeth.

Among bovid mammals, the evolution of female weaponry has been closely linked to defense, particularly in open habitats in which crypsis is not effective [66] (Stankowich & Caro, 2009). Both sexes of †*O. rastrosus* would have been similarly vulnerable to predation during the open ocean or return phases of their semelparous anadromous migrations. Modernly, orcas feed preferentially upon King Salmon (*O. tshawytscha*) [67] (Vester & Hammerschmidt, 2013), sea lions often feast upon salmonids congregating at river mouths [68] (Roffe & Mate, 1984), and bears consume many individuals during their upstream migrations, particularly at difficult waterfall passages [69] (Levi, et al., 2020). Given the considerable lateral force that the axial musculature of a two-meter-long fish could impart, we hypothesize that †*O. rastrosus* could have easily killed or injured its likely predators with the projecting tusks.

Overall, the blunt morphology of the premaxillary teeth, their lateral direction, and the probable dominance of a filter feeding strategy indicate that †*Oncorhynchus rastrosus* did not use its teeth as offensive weapons. However, a defensive function remains plausible.

**Competition.** Intraspecific agonism regularly drives the evolution of animal weaponry and ornaments, most frequently leading to sexual dimorphism and augmented morphology of whichever sex competes for access to the other, which is usually the male [70] (Peterson, Dischler, & Longrich, 2013). Salmonids certainly demonstrate sexual dimorphism and agonistic behavior, with males developing hooked kypes, humps and enlarged teeth during the breeding season and aggressively competing for breeding sites and access to females [28,71] (Vladykov, 1962; Fleming & Reynolds, 2004). If the presence of the enlarged teeth were sexually dimorphic in †*Oncorhynchus rastrosus*, then their use in intraspecific competition would be the leading explanation for hypertrophy. However, the teeth are monomorphic (Fig 4A and 4B). While sexual monomorphism does not rule out agonism, it does warrant further scrutiny. Female weaponry may arise indirectly through shared genetic architecture with a positively selected male trait or might be selected directly [72] (Tobias, Montgomerie, & Lyon, 2012). In the latter case, the weapons might aid males and females in competition for limited ecological resources, or they might arise from sexually mediated competition among the females themselves. Without the ability to observe the behavior of this extinct species it is difficult to move past conjecture, but it does seem reasonable to conclude that †*O. rastrosus* retained the high levels of agonism during breeding that characterize its genus generally. If so, the shared possession of the enlarged teeth by males and females raises the intriguing idea that both sexes may have competed for limited resources, such as access to the best sites for redds or the opportunity to mate with dominant individuals of the other sex.

**Tools.** The spikes of this salmon might have served as practical implements to excavate redd sites. While true tool use is rare among actinopterygians and mostly limited to labrids

[73] (Bernardi, 2012), numerous examples exist of ray-finned fishes using their bodies to manipulate their environment. For example, some cichlids engage in fin-digging behavior to help provision their offspring [74] (Zworykin, Budaev, & Mochek, 2000), while gobies [75] (Choi & Gushima, 2002) and flying gurnards [76] (Davenport & Wirtz, 2019) dig through the substrate to locate high quality food for themselves. Though salmonids typically use their tails and strong body musculature to excavate the redds into which they lay their eggs [77] (Burner, 1951), many other fishes use their mouths to excavate or construct nests such as minnows of the genus *Semotilus* [78] (Maurakis, Woolcott, & Magee, 1990), various cichlids [79] (Lowe, 1956) and lampreys [80] (Hagelin & Steffner, 1958). †*Oncorhynchus rastrosus* could have plausibly used its lateral spikes to aid in redd construction. Such a huge salmonid species would have required quite a large redd, possibly one deeper than needed by its smaller congeners, and the tail alone was potentially insufficient to fully excavate the optimal nests.

The premaxillary tooth is also sharper in coastal marine specimens. The specimens are all found in sediments deposited in large and, at times, fast-flowing rivers, which is relevant to the potential wear pattern seen on teeth. If the wear patterns resulted from post-mortem specimens, it would be more likely that specimens found more downstream will show more wear due to further stream transport. While the bone was worn in both coastal marine and upstream freshwater localities, the teeth were sharpest in the former and dullest in the latter. Therefore, the greater wear of premaxillary teeth in large adults from freshwater environments provides evidence for use by males and females alike during the final phase of their life history.

**Multifunctionality.**   The hypertrophied premaxilla and lateral spikes of †*Oncorhynchus rastrosus* could have served several plausible functions, including defense against predators, agonism against conspecific individuals, or as a practical aid to nest construction. These functions are not mutually exclusive, and all may have come into play occasionally. Similar multifunctionality can be seen in the remarkable rostrum of sawfishes, which can function offensively to impale prey [81,82] (Breder, 1952; Wueringer, Squire, & Collin, The biology of extinct and extant sawfish (Batoidea: Sclerorhynchidae and Pristidae), 2009) and also serves to enhance the animal's electrosensory capacities [83] (Wueringer, Electroreception in elasmobranchs: sawfish as a case study, 2012). Among other cartilaginous fishes, such as all species of Squatiniformes and Pristiophoriformes, as well as some members of Orectolobiformes and Carcharhiniformes, rostra are expanded and basiventral cartilages are laterally expanded, contributing to a broad articulation with the occipital condyle [84] (Claeson & Hilger, 2011). A similar phenomenon is noted at the craniovertebral junction of †*Oncorhynchus rastrosus* (Fig 7A and 7B), suggesting a significant functional signal, likely related to the need for cranial stability during lateral undulation when using said spikes. Given the lateral orientation of this enormous salmon's teeth and its assumed zooplanktivory, we consider predation unlikely as the primary tooth function, but most other explanations of why this greatest of all salmonids evolved its spikes remain open.

## Conclusion

Extant Pacific salmon are sexually dimorphic, anadromous fishes navigating complex terrains to successfully breed. They undergo impressive physiological and morphological transformations prior to and during migration inland to spawn. †*Oncorhynchus rastrosus* was no exception, despite its huge size and unusual filter feeding ecology. The newly recovered specimens from the Gateway Locality of Oregon represent mature individuals with subtle dimorphism in the vomer, rostro-dermethmoid-supraethmoid, and dentary. Male specimens possess a more elongate vomer than do females and do not have a dorsal keel as in females. The female dentary has no evidence of a kype and possesses two mesial teeth. Male specimens of †*O. rastrosus*

have a non-uniform density deep to the tooth bed and edentulous portion of the dentary, which we interpret as evidence of a kype. Finally, the rds is blunt in females and pointed in males. Unlike extant *Oncorhynchus*, male and female specimens of †*O. rastrosus* do not differ in premaxilla shape and both sexes possess prominent, laterally-directed premaxillary teeth. These spikes on the hypertrophied premaxilla could have defended against predators, enhanced agonism against conspecific individuals, and/or aided nest construction.

## Supporting information

**S1 Video. Sagittal cross sectional CT scan data of UOF_55101A.**
(MP4)

**S1 Matrix. Character matrix with Bayesian parameters.**
(TXT)

**S1 Fig. Cover art of female salmon.**
(TIF)

## Acknowledgments

We thank Gateway Quarry Landowners for access to the land and donation of specimens. G. Carr and North American Research Group for donating specimens; P. Konstantinidis and G. R. Smith provided helpful discussion and insight about premaxillary tooth morphology among fishes. The Academy of Natural Sciences Philadelphia of Drexel University provided access to VGStudio Max.

## Author Contributions

**Conceptualization:** Kerin M. Claeson, Brian L. Sidlauskas, Edward B. Davis.

**Data curation:** Kerin M. Claeson, Ray Troll, Edward B. Davis.

**Formal analysis:** Kerin M. Claeson, Zabrina M. Prescott.

**Funding acquisition:** Kerin M. Claeson, Edward B. Davis.

**Investigation:** Kerin M. Claeson, Brian L. Sidlauskas, Zabrina M. Prescott.

**Methodology:** Kerin M. Claeson, Brian L. Sidlauskas, Zabrina M. Prescott.

**Project administration:** Kerin M. Claeson, Edward B. Davis.

**Resources:** Kerin M. Claeson, Edward B. Davis.

**Software:** Kerin M. Claeson.

**Supervision:** Kerin M. Claeson, Edward B. Davis.

**Validation:** Kerin M. Claeson.

**Visualization:** Kerin M. Claeson, Ray Troll.

**Writing – original draft:** Kerin M. Claeson, Brian L. Sidlauskas, Ray Troll, Zabrina M. Prescott, Edward B. Davis.

**Writing – review & editing:** Kerin M. Claeson, Brian L. Sidlauskas, Ray Troll, Zabrina M. Prescott, Edward B. Davis.

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
