## [Decision Letter · Decision Letter 0]

30 Nov 2023

PONE-D-23-29788From sabers to spikes: a newfangled reconstruction of the ancient, giant, sexually dimorphic Pacific salmon, †*Oncorhynchus rastrosus*(SALMONINAE: SALMONINI)PLOS ONE

Dear Dr. Claeson,

Thank you for submitting your manuscript to PLOS ONE. After careful consideration, we feel that it has merit but does not fully meet PLOS ONE’s publication criteria as it currently stands. Therefore, we invite you to submit a revised version of the manuscript that addresses the points raised during the review process.

The reviewers are very pleased with the manuscript, only minor corrections have been suggested. Reviewer 2 remarks that the "The extended discussion of possible functions seems to me to be a bit over the top in terms of detail, and the language used is rather informal". I agree with this remark and thus have inserted a few detailed comments on the discussion. If you can take care of the minor comments there should be no problem to publish this very interesting manuscript. I am looking forward to the revised version.

We look forward to receiving your revised manuscript.

Kind regards,

Paul Eckhard Witten, PhD

Academic Editor

PLOS ONE

2. In your manuscript, please provide additional information regarding the specimens used in your study. Ensure that you have reported human remain specimen numbers and complete repository information, including museum name and geographic location.

For more information on PLOS ONE's requirements for paleontology and archeology research, see https://journals.plos.org/plosone/s/submission-guidelines#loc-paleontology-and-archaeology-research.

4. We note that Figures 1, 2, 3, 4, 5, 6, 7 ,8, 9 and S1Fig.tif in your submission contain copyrighted images. All PLOS content is published under the Creative Commons Attribution License (CC BY 4.0), which means that the manuscript, images, and Supporting Information files will be freely available online, and any third party is permitted to access, download, copy, distribute, and use these materials in any way, even commercially, with proper attribution. For more information, see our copyright guidelines: http://journals.plos.org/plosone/s/licenses-and-copyright.

1. You may seek permission from the original copyright holder of Figures 1, 2, 3, 4, 5, 6, 7 ,8, 9 and S1Fig.tif to publish the content specifically under the CC BY 4.0 license.

5. We are unable to open your Supporting Information file [S1Matrix.nex]. Please kindly revise as necessary and re-upload.

Additional Editor Comments:

Dear Dr. Claeson,

the reviewers are very pleased with the manuscript, only minor corrections have been suggested. Reviewer 2 remarks that the "The extended discussion of possible functions seems to me to be a bit over the top in terms of detail, and the language used is rather informal". I agree with this remark and thus have inserted a few detailed comments on the discussion. If you can take care of the comments, all of which are minor, there should be no problem to publish this very interesting manuscript. I am looking forward to the revised version.

Kind regards, Eckhard Witten

Reviewer1:

This is an extremely interesting article for several reasons. The material is new, well preserved, and phylogenetically informative. The use of tomographic scanning has opened up a wealth of useful and interesting data, which the authors have put fully to use! Finally (and most impreesively), the paper showcases the scientific value of a wide collaboration between collectors, scientists with access to sophisticated preparation and scanning technology, and a world-renowned scientific artist, Congratulations to all

I have no criticisms of the paper. It has been well written, beautifully illustrated, and is one of the most comprehensive and thorough scientific investigations of any fossil material I have seen.

Reviewer 2:

This is a fine and detailed reassessment of an amazing fish species. The techniques are appropriate. The descriptive anatomy is good and the illustrations are necessary and appropriate. I have only minor suggestions.

1. I am not sure what the authors mean by the term "salmonins." Their systematic section classifies the genus within the subfamily Salmoninae, but the correct/standard way to informally refer to that clade is "salmonines"

If they mean to refer to a tribe or subtribe, that would require yet another spelling.

2. The extended discussion of possible functions seems to me to be a bit over the top in terms of detail, and the language used is rather informal compared to the rest of the manuscript. I think it could be shortened by about a quarter and revised to a more technical style. I also noticed that some but not all of the in-text citations use the author-date style whereas most of the manuscript uses numbered citations.

3. Throughout my reading I noticed occasional problems of punctuation (missing or incorrect) and rarely missing words. Mostly it's well written. I left it to the authors and copy editor to find and fix those in the final draft.

4. I am sceptical of the idea that these huge fishes were semelparous, even though their relatives were and are. I can't think of many examples of huge animals that breed only once. Is there any way to determine (growth rings on vertebrae, otoliths, ?) how old they were when they ventured into and died in freshwater environments, and/or how long they had to live in the ocean to reach their final sizes? Are all the freshwater fossil finds of a similar huge size or might there be different ages/sizes? I am thinking that these fish were far larger than necessary to make the return trip to the lakes and rivers. Probably they were too large to breed in the shallows of smaller rivers and streams.

Overall, I like this paper.

Editors comments:

Line 610: please add a reference

Line 626-643, paragraph, discussion on the kype: Please add the published peer revived literature on the subject

Line 738: "fishes" is perhaps not a good term in a scientific paper. Comparing the function of enlarged teeth in groups as distant as elasmobranchs and actinopterygians one could as well discuss elephant tusks. At least elephants and salmonids are both in the same group, osteichthyans.

Line 755, please add species name or insert spp. if applies to the entire genus or sp. if the species is unknown

Line 790: please replace "fishes" by a specific scientific term

Line 790-792: axial muscles for raid moth extension, please check if this is correct

Line 824 and 828: please replace "fishes" by a specific scientific term

Line 841: spikes or teeth?

Line 853: add assumed, "assumed zooplanctivory…"

Line 853: "we can reliably rule out" this is a very strong statement, perhaps better down tune a bit

Line 876-868: …"edentulous portion of the dentary, which we interpret as evidence of a kype". Please consider, (a) in Atlantic salmon can have teeth on the kype, (b) in Atlantic salmon teeth are ankylosed to the jaw but the connection to the jaw is not fully mineralised. This was the reason that, based on macerated specimens, an edentulous stage prior to breeding has been described This turned out turned out not to be true, a maceration artefact. There is detailed published literature about the kype and the dentition of Atlantic salmon.

Line 871:…"plausible multifunctional". This is speculation that and does not add to the scientific discussion.

Reviewers' comments:

Reviewer's Responses to Questions

**Comments to the Author**

1. Is the manuscript technically sound, and do the data support the conclusions?

Reviewer #1: Yes

Reviewer #2: Yes

2. Has the statistical analysis been performed appropriately and rigorously? 

Reviewer #1: Yes

Reviewer #2: N/A

3. Have the authors made all data underlying the findings in their manuscript fully available?

Reviewer #1: Yes

Reviewer #2: Yes

4. Is the manuscript presented in an intelligible fashion and written in standard English?

Reviewer #1: Yes

Reviewer #2: Yes

5. Review Comments to the Author

Reviewer #1: This is an extremely interesting article for several reasons. The material is new, well preserved, and phylogenetically informative. The use of tomographic scanning has opened up a wealth of useful and interesting data, which the authors have put fully to use! Finally (and most impreesively), the paper showcases the scientific value of a wide collaboration between collectors, scientists with access to sophisticated preparation and scanning technology, and a world-renowned scientific artist, Congratulations to all.

Reviewer #2: This is a fine and detailed reassessment of an amazing fish species. The techniques are appropriate. The descriptive anatomy is good and the illustrations are necessary and appropriate. I have only minor suggestions.

1. I am not sure what the authors mean by the term "salmonins." Their systematic section classifies the genus within the subfamily Salmoninae, but the correct/standard way to informally refer to that clade is "salmonines"

If they mean to refer to a tribe or subtribe, that would require yet another spelling.

2. The extended discussion of possible functions seems to me to be a bit over the top in terms of detail, and the language used is rather informal compared to the rest of the manuscript. I think it could be shortened by about a quarter and revised to a more technical style. I also noticed that some but not all of the in-text ciitations use the author-date style whereas most of the manuscript uses numbered citations.

3. Throughout my reading I noticed occasional problems of punctuation (missing or incorrect) and rarely missing words. Mostly it's well written. I left it to the authors and copy editor to find and fix those in the final draft.

4. I am skeptical of the idea that these huge fishes were semelparous, even though their relatives were and are. I can't think of many examples of huge animals that breed only once. Is there any way to determine (growth rings on vertebrae, otoliths, ?) how old they were when they ventured into and died in freshwater environments, and/or how long they had to live in the ocean to reach their final sizes? Are all the freshwater fossil finds of a similar huge size or might there be different ages/sizes? I am thinking that these fish were far larger than necessary to make the return trip to the lakes and rivers. Probably they were too large to breed in the shallows of smaller rivers and streams.

Overall, I like this paper.

6. PLOS authors have the option to publish the peer review history of their article (what does this mean?). If published, this will include your full peer review and any attached files.

Reviewer #1: **Yes: **John G, Maisey, Curator Emeritus, American Museum of Natural History, NY, USA.

Reviewer #2: No

---

## [Author Response · Author response to Decision Letter 0]

23 Jan 2024

Response to reviewers and editor comments: 

To Reviewer1: Thank you. We are excited to share these outcomes with the broader scientific and public communities. 

Reviewer 2:

This is a fine and detailed reassessment of an amazing fish species. The techniques are appropriate. The descriptive anatomy is good and the illustrations are necessary and appropriate. I have only minor suggestions.

1. I am not sure what the authors mean by the term "salmonins." Their systematic section classifies the genus within the subfamily Salmoninae, but the correct/standard way to informally refer to that clade is "salmonines"

If they mean to refer to a tribe or subtribe, that would require yet another spelling.

Response: These instances were corrected to salmonine(s).

2. The extended discussion of possible functions seems to me to be a bit over the top in terms of detail, and the language used is rather informal compared to the rest of the manuscript. I think it could be shortened by about a quarter and revised to a more technical style. I also noticed that some but not all of the in-text citations use the author-date style whereas most of the manuscript uses numbered citations.

Response: A more technical style was implemented and as a result, we reduced the text in that section by nearly a quarter. The author-date style citations were corrected to follow the Vancouver format. 

3. Throughout my reading I noticed occasional problems of punctuation (missing or incorrect) and rarely missing words. Mostly it's well written. I left it to the authors and copy editor to find and fix those in the final draft.

Response: We each read the manuscript again to find as many of these instances as possible. 

4. I am sceptical of the idea that these huge fishes were semelparous, even though their relatives were and are. I can't think of many examples of huge animals that breed only once. Is there any way to determine (growth rings on vertebrae, otoliths, ?) how old they were when they ventured into and died in freshwater environments, and/or how long they had to live in the ocean to reach their final sizes? Are all the freshwater fossil finds of a similar huge size or might there be different ages/sizes? I am thinking that these fish were far larger than necessary to make the return trip to the lakes and rivers. Probably they were too large to breed in the shallows of smaller rivers and streams.

Response: We reduced this section and will revisit all of these questions as part of the subject of a future manuscript.

Overall, I like this paper.

Thank you!

Editors comments:

Line 610: please add a reference - added

Line 626-643, paragraph, discussion on the kype: Please add the published peer revived literature on the subject - Section revised and removed

Line 738: "fishes" is perhaps not a good term in a scientific paper. Comparing the function of enlarged teeth in groups as distant as elasmobranchs and actinopterygians one could as well discuss elephant tusks. At least elephants and salmonids are both in the same group, osteichthyans. - Noted and revised as “aquatic vertebrates” 

Line 755, please add species name or insert spp. if applies to the entire genus or sp. if the species is unknown - added

Line 790: please replace "fishes" by a specific scientific term - revised as “actinopterygian and chondrichthyan fishes”

Line 790-792: axial muscles for raid moth extension, please check if this is correct - Yes, in a study on large mouth bass, axial muscles “were the primary source of suction expansion power and generated up to 95% of peak expansion power.” Camp et al. (2015) 

Line 824 and 828: please replace "fishes" by a specific scientific term - “aquatic vertebrates” and “actinopterygians”

Line 841: spikes or teeth? - “hypertrophied premaxilla and lateral spikes”

Line 853: add assumed, "assumed zooplanctivory…" - added

Line 853: "we can reliably rule out" this is a very strong statement, perhaps better down tune a bit - adjusted

Line 876-868: …"edentulous portion of the dentary, which we interpret as evidence of a kype". Please consider, (a) in Atlantic salmon can have teeth on the kype, (b) in Atlantic salmon teeth are ankylosed to the jaw but the connection to the jaw is not fully mineralised. This was the reason that, based on macerated specimens, an edentulous stage prior to breeding has been described This turned out turned out not to be true, a maceration artefact. There is detailed published literature about the kype and the dentition of Atlantic salmon. - Understood. The concluding statement remains the same, however, earlier discussion is revised. 

Line 871:…"plausible multifunctional". This is speculation that and does not add to the scientific discussion. - Revised

Response: All files were updated. 

2. In your manuscript, please provide additional information regarding the specimens used in your study. Ensure that you have reported human remain specimen numbers and complete repository information, including museum name and geographic location.

Responses: The section on specimens examined was expanded to include specimens numbers for comparative materials. Acknowledgment for permission to collect on private lands is provided in the text. A statement indicating no permits were required is added to the section on specimens examined. Locality and age were added to section on Systematic paleontology

3. We note that you have stated that you will provide repository information for your data at acceptance.

Response: The section on specimens examined was expanded to include specimens numbers for comparative materials and the abbreviates section was expanded to include all institutions from which specimens were examined. 

4. We note that Figures 1, 2, 3, 4, 5, 6, 7 ,8, 9 and S1Fig.tif in your submission contain copyrighted images. 

Response: Co-Author Ray Troll has provided permission to publish his artwork (Strike image and Fig. 1C-D). All other figures are original artwork, CT modeling, and photography by co-author, Claeson. 

5. We are unable to open your Supporting Information file [S1Matrix.nex]. Please kindly revise as necessary and re-upload.

Response: File is uploaded as a simple text file instead of a nexus file. The same information will be downloadable as a nexus from MorphoBank.org upon publication should future researchers wish to avoid transcribing the dataset themselves. 

6. Please review your reference list to ensure that it is complete and correct.

Response: Updates were made to citations included in the manuscript as needed when text was modified. The number changes are reflected in the text and can be seen in track changes. No citations were retracted.

---

## [Decision Letter · Decision Letter 1]

26 Feb 2024

From sabers to spikes: a newfangled reconstruction of the ancient, giant, sexually dimorphic Pacific salmon, †Oncorhynchus rastrosus (SALMONINAE: SALMONINI)

PONE-D-23-29788R1

Dear Dr.  Claeson,

We’re pleased to inform you that your manuscript has been judged scientifically suitable for publication and will be formally accepted for publication once it meets all outstanding technical requirements.

Kind regards,

Paul Eckhard Witten, PhD

Academic Editor

PLOS ONE

Additional Editor Comments (optional):

Reviewers' comments:

Reviewer's Responses to Questions

**Comments to the Author**

1. If the authors have adequately addressed your comments raised in a previous round of review and you feel that this manuscript is now acceptable for publication, you may indicate that here to bypass the “Comments to the Author” section, enter your conflict of interest statement in the “Confidential to Editor” section, and submit your "Accept" recommendation.

Reviewer #1: All comments have been addressed

Reviewer #2: All comments have been addressed

2. Is the manuscript technically sound, and do the data support the conclusions?

Reviewer #1: Yes

Reviewer #2: Yes

3. Has the statistical analysis been performed appropriately and rigorously? 

Reviewer #1: Yes

Reviewer #2: Yes

4. Have the authors made all data underlying the findings in their manuscript fully available?

Reviewer #1: Yes

Reviewer #2: Yes

5. Is the manuscript presented in an intelligible fashion and written in standard English?

Reviewer #1: Yes

Reviewer #2: Yes

6. Review Comments to the Author

Reviewer #1: I have no substantial criticisms. As far as I'm concerned, the paper is now ready for publication.

Reviewer #2: I am satisfied with the revisions. The paper is even more interesting in revised form. I will look forward to seeing it in published form and also any further studies undertaken about its life history.

7. PLOS authors have the option to publish the peer review history of their article (what does this mean?). If published, this will include your full peer review and any attached files.

Reviewer #1: No

Reviewer #2: **Yes: **Mark V. H. Wilson

---

## [Editor Report · Acceptance letter]

1 Apr 2024

PONE-D-23-29788R1 

PLOS ONE

Dear Dr. Claeson, 

I'm pleased to inform you that your manuscript has been deemed suitable for publication in PLOS ONE. Congratulations! Your manuscript is now being handed over to our production team.

Kind regards, 

on behalf of

Dr. Paul Eckhard Witten 

Academic Editor

PLOS ONE